

# Flood-pedestrian simulator for modelling human response dynamics during flood-induced evacuation: Hillsborough stadium case study

Mohammad Shirvani[1], Georges Kesserwani[1]

[1]Department of Civil and Structural Engineering, University of Sheffield, Mappin St, Sheffield City Centre, Sheffield S1 3JD, UK

*Correspondence to*: Georges Kesserwani (g.kesserwani@sheffield.ac.uk)

**Abstract.** The flood-pedestrian simulator uses a parallel approach to couple a hydrodynamic model to a pedestrian model in a single agent-based modelling (ABM) framework using the Graphical Processing Units (GPU). It allows to dynamically exchange and process multiple agent information between the two models. The simulator is augmented with more realistic human-body characteristics and in-model behavioural rules. The new features are implemented in the pedestrian model to factor in age- and gender-related walking speeds for the pedestrians in dry zones with possible acceleration based on a maximum excitement condition. It is also adapted to use age-related moving speeds for pedestrians inside flooded zones, with either a walking condition or a running condition. The walking and running conditions are applicable without and with an existing two-way interaction condition that considers the effects of pedestrian congestion on the floodwater dynamics. A new autonomous change of direction condition is proposed to make way-finding decisions of pedestrians self-automated, based on the changes in the state of the floodwater dynamics or the most popular choice of destination selected by the other pedestrians. The relevance of the newly added characteristics and rules is demonstrated by applying the augmented simulator to reproduce a flood evacuation in a shopping centre test case, which was investigated with the version of the simulator that does not consider age and gender characteristics in the pedestrian model. The augmented simulator is finally applied to study evacuation scenarios for a mass evacuation from the Hillsborough football stadium located in a flood-prone area in Sheffield.

## 1 Introduction

Flooding can disturb local communities such as in and around urban hubs, putting people at risk (Flood and coastal erosion risk management policy statement, 2020). In lead-up to, and during, an urban flood people could be caught at different flood risk states depending on their body characteristics, individual actions, and interaction with the floodwater and each other (Environment Agency, 2006; Milanesi et al., 2015; Arrighi et al., 2017; Musolino et al., 2020; Moftakhari et al. 2018; Rufat et al., 2020; Hamilton et al., 2020). Understanding and quantifying how the interplay between people-related aspects affect flood risk to people is a desired way forward (Aerts et al., 2018). In the context of the flood risk management, methods and computational models have been developed to incorporate one or more of these aspects. This has been done with a particular



focus on assessing and minimising the direct risks of floodwater on people, especially when they are under immediate
evacuation condition. For this purpose, computer models have been used to evaluate evacuation strategies, including for the
analysis of the potential variability in individual people's flood risk state, the lowest-risk pathways, and the time window for
issuing an early warning (Aboelata and Bowloes, 2008; Lumbroso et al., 2011; Dawson et al., 2011; Mas et al., 2015; Liu and
Lim, 2016; Bernardini et al., 2017; Zhu et al., 2019).
Most of the existing evacuation models are built upon the soft agent-based modelling (ABM) paradigm, as it offers
the flexibility needed to represent both the interactive and the collective responses of people, either considered as moving
individuals, groups of individuals in a vehicle, or household units (Zhuo and Han, 2020; Aerts, 2020). Incorporating these
responses into ABM-based models enables to simulate people response dynamics into the flood risk analysis, leading to more
informed predictions for flood adaptation planning and extraction of decision-relevant indicators (Aerts et al., 2018;
McClymont et al., 2019). A two-dimensional hydrodynamic model is often used to feed information on the extent of the
floodwater depth and velocity magnitude, as inputs, into ABM-based evacuation models from which the interactions between
people and the floodwater could be modelled (Dawson et al., 2011; Bernardini et al., 2017; Aerts, 2020). These interactions
directly affect the moving speed and stability states of people in and around the floodwater as they respond to an emergency
warning while interacting with the features of an urban layout (Shirvani et al. 2020). The representation of these interactions
within ABM-based models seems to require different levels of sophistication for the agent characteristics and behavioural rules
depending on the purpose of the model design and the targeted scale of application.
For regional scale applications, ABM-based models were developed to simulate immediate crowd evacuation from a
city, focusing on moving groups of individuals or household units using cars within a city road network to analyse response
time of aware and unaware people to the immediate evacuation warning (Dawson et al., 2011; Mas et al., 2015; Liu and Lim,
2016; Zhu et al., 2019). These simulation models only consider vehicular emergency evacuation, which makes them not suited
to simulate the interactive and the collective responses of moving individuals, or pedestrians, in and around small hubs (< 0.5
km × 0.5 km in size), such as shopping centres or sports venues. For such local scale applications, where pedestrians need to
be individually modeled, only a few ABM-based evacuation models were reported. One of these models is the Life Safety
Model (LSM, www.lifesafetymodel.net) developed by BC Hydro and HR Wallingford, which allows to analyse evacuation
patterns of pedestrians along streetscapes and crossings (Lumbroso and Di Mauro, 2008; Lumbroso and Davison, 2018).
Another model is LifeSIM (www.hec.usace.army.mil/software/hec-lifesim), developed by the US Army Corps of Engineers,
which is capable of simulating individuals' responses to an emergency warning with the floodwater propagation, as they
interact with the features of an urban layout, e.g. streetscapes and buildings (Aboelata and Bowles, 2008). These ABM-based
evacuation models were developed to estimate the number of casualties and injuries and evacuation time, and to pinpoint
highest-risk locations due to severe flood types, e.g. in the immediate aftermath of a dam-break or a tsunami wave where
individual's decisions and actions have insignificant influence on the model outcomes. Less attention has been given to model
the microscopic responses, down to the scale of the moving individuals, in and around flooded urban hubs for the most common
flood types that would often lead to a low to moderate flood risk to people (Environment Agency, 2006).



One first step to designing an ABM-based evacuation model capable of capturing microscopic responses at a small

urban scale was taken by Bernardini et al. (2017), who developed FlooPEDS. In FlooPEDS, the standard social force model

for pedestrian dynamics (Helbing and Molnar, 1995) was adapted to further model individuals' moving speed and stability

states in floodwater. These states were implemented based on the experimental data and recommendations in Ishigaki et al.

(2009), Chanson et al. (2014) and Matsuo et al. (2011), though individuals' way-finding decisions were solely influenced by

behavioural rules of the social force model. The coupling with the hydrodynamic model was used to receive information on

the changes in the floodwater conditions within the urban environment. However, FlooPEDS was reported to adopt a serial

approach, by running one of the social force model and hydrodynamic model at a time, and a number of simplifications to

alleviate runtime and dynamic memory costs, i.e. using uniform floodwater conditions on coarse subdomains, limiting the

number of pedestrians up to 300 with uniform characteristics and the simulation time to less than 600 s (Bernardini et al.,

2017). Given its serial approach to the coupling, FlooPEDS is not ideally suited to incorporate the dynamic feedback from the

moving pedestrians onto the floodwater flow. Shirvani et al. (2021) developed a flood-pedestrian simulator by taking a parallel

approach to achieve the dynamic coupling between the hydrodynamic model and the social force model, both being ABM-

based and running from a single ABM-based framework, Flexible Large-scale Agent-based Modelling Environment for the

GPU (FLAMEGPU). The flood-pedestrian simulator on the FLAMEGPU framework benefits from the computational speed-

up and high dynamic memory capacity of the Graphics Processing Unit (GPU). The latter property allows it to employ as fine

resolution and as large population size as needed with the hydrodynamic and pedestrian models, respectively (within the

capacity of available GPU memory). This simulator is therefore supported with a two-way interaction condition to dynamically

exchange agent information as they get updated across both the social force model and the hydrodynamic model. The two-

way interaction condition allows to capture both the response of moving pedestrians to the floodwater and the back interaction

of pedestrians' presence on the floodwater flow. Enabling the two-way interaction condition was found to significantly affect

the model outcomes in and around congested areas: predict reduced flood risk for the pedestrians in low to medium risk areas

and increased risk for those around high risk areas (Shirvani et al., 2021). In Shirvani et al. (2020), the social force model of

the same simulator was further augmented with empirical datasets and experimentally derived formulas to incorporate non-

uniform body characteristics and more variable moving speed and stability states of pedestrian agents in floodwater. The

simulator was found to predict significantly prolonged evacuation times and higher number of at-risk pedestrians in low to

medium risk areas in line with an increased sophistication in the pedestrian agent characteristics and behavioural rules (Shirvani

et al., 2020), even without enabling the two-way interaction condition. In the latter version of the simulator, pedestrian agents

were initialised to store body height and mass information, which were the only human body factors considered to influence

the determination of their stability states in the floodwater; and, were assigned variable moving speeds that are solely based

on the mechanics of the floodwater. Also, the latter version of the simulator was limited to assigning a way-finding decision

for the pedestrian agents based on the availability of a fixed emergency exit destination (specified in advance), and did not

explore the interplay between the two-way interaction condition and the pedestrian agent characteristics and rules.





This paper presents the latest version of the simulator, augmented with new characteristics and rules for the pedestrian
agents to more realistically represent the behaviour of people around and inside urban floodwater. In this version, pedestrian
agents are initialised to store information of age, gender and more realistic body mass that are all accounted for within the rules
determining both the moving speed and the stability states of the pedestrian agents inside the floodwater. The social force
model is supplemented with empirically based age- and gender-related variable walking speeds to model the response of
pedestrians in dry zones around the floodwater,  and a new rule to incorporate a maximum excitement condition for replicating
a faster than normal moving speed under evacuation condition. It is also supplemented with an experimentally derived formula
to dynamically adapt moving speeds of pedestrian agents inside the floodwater that are also based on their age and gender, to
include either a walking condition or a running condition. Each of these added rules could be applied with and without the
two-way interaction condition. A new autonomous change of direction condition is proposed to enable the pedestrian agents
to automatically make their way-finding decision informed by the state of the floodwater or the choices of pathways selected
by their neighbouring agents. The added characteristics and behavioural rules are evaluated by analysing the associated changes
induced in the simulated outcomes and also referring to the outcomes obtained from the latter version of the simulator. The
utility of the simulator, with the new autonomous change of direction condition, is demonstrated to study evacuation scenarios
for a mass evacuation from the Hillsborough football stadium in response to a flood emergency replicating the November 2019
Sheffield floods. Discussions of key findings and limitations are finally presented. Because the flood-pedestrian simulator is
flexibly configurable to adopt other case studies and evacuation scenarios, its source code on the FLAMEGPU framework,
including all the agent characteristics and/or rules, is made openly available (Shirvani and Kesserwani, 2021a), as well as the
datasets from the simulated case studies (Shirvani and Kesserwani, 2021b) and a video supplement that visualises the
simulations in real time (Shirvani, 2021).
**2 Material and methods**
**2.1 Overview of the flood-pedestrian simulator**
The flood-pedestrian simulator dynamically couples a hydrodynamic model to a pedestrian model within the same agent-based
modelling (ABM) framework, FLAMEGPU (Shirvani et al., 2020; Shirvani et al., 2021). The pedestrian model adopts a
standard social force model that accounts for the dynamic interactions occurring between moving pedestrians in a built
environment (Li et al., 2019; Jiang et al., 2020). The pedestrians are represented by continuous space agents, each of which
autonomously move in space and over time. The movement pattern of each pedestrian agent is derived by forces for avoiding
collisions with their neighbouring pedestrian agents and with the key features within the environment layout, such as
boundaries of the walkable area, terrain blocks and solid walls. The environment layout encodes force vector fields providing
navigation to key destinations. These fields are stored within a grid of fixed discrete agents, forming a navigation map
(Karmakharm et al., 2010). The navigation map is necessary for pedestrians' way-finding decisions while they are directed to
reach their key destinations.



The hydrodynamic model is applied on another fixed grid of discrete agents, flood agents, which is coincident with
the grid of navigation agents. A flood agent stores information of the terrain properties in terms of height ($z$) and Manning's
roughness parameter ($n_M$); and the state of floodwater variables in terms of depth ($h$) and velocity components ($u$ and $v$). The
state of floodwater variables is updated over time using a non-sequential computation of a hydrodynamic model that operates
for all the flood agents at the same time. Each navigation agent thereby stores the updated state of floodwater variables from
the coincident flood agent and translates it into a local flood hazard rating (HR) quantity. This HR quantity is estimated as HR
$= (V + 0.5) \times h$ where $V$ stands for the velocity magnitude, estimated as $V = \sqrt{u^2 + v^2}$, (Environment Agency, 2006). This HR
quantity is usually used as a metric to assess the degree of flood risk to people (Kvočka et al., 2016; Willis et al., 2019;
Costabile et al., 2020). Therefore, HR-related flood risk states are used based on the categorisation of flood risk to people
reported in the UK Environment Agency (EA), as listed in Table 1.

**Table 1:** The flood risk states assigned for the pedestrian agents according to the HR ranges reported by the Environment Agency (2006)

| HR range | Flood risk state |
|---|---|
| 0.00 to 0.75 | 'Low' (safe for all) |
| 0.75 to 1.50 | 'Medium' (danger for some, i.e. children) |
| 1.50 to 2.50 | 'High' (danger for most) |
| 2.50 to 20.0 | 'highest' (danger for all) |


The navigation agents are set to act as a shared communication interface between the flood agents and pedestrian
agents, where the information is dynamically exchanged between the three agents (Shirvani et al., 2021). This interaction
mechanism allows a pedestrian agent to pick up a HR quantity from the navigation agent at its location and autonomously flag
itself with one of the flood risk states listed in Table 1. Also, pedestrian agents are enabled to adapt their moving speed based
on the change in the state of floodwater variables accessible from the navigation agents at its location and time.
The simulator is also supported with a functionality to enable a 'two-way interaction condition' to consider the effects
that pedestrians' congestion would have on altering the floodwater hydrodynamics, which can be significant as shown in
Arrighi et al. (2017) and Shirvani et al. (2021). Hence, this condition incorporates any local and temporal changes in the states
of the floodwater variables in a flood agent as a result of increased accumulation of pedestrian agents over the navigation agent
at its coincident location. By enabling this functionality, the navigation agent is set to count the number of pedestrian agents
($N_p$) that occupy its area at each time step. Then, the navigation agent uses $N_p$ to alter local energy loss by locally updating $n_M$
and passing it back to the coincident flood agent. The updated $n_M$ is applied as $n_M^{updated} = n_M + (N_p \times n_M)$. The initial $n_M$
parameter is set to be equal to $0.01$ s.m$^{-1/3}$, representative of clear cement, and no more than 20 pedestrian agents are allowed
to simultaneously occupy the area of a navigation agent, meaning that any local update in $n_M$ cannot exceed $0.2$ m$^{-1/3}$.
In the previous version of the simulator (Shirvani et al., 2020), the variable moving speed of pedestrian agents in
floodwater was locally estimated based on an empirical formula that only considers the mechanics of floodwater depth and





velocities (Bernardini et al., 2017). This formula does not take into account the impact of different age and gender groups on the moving speeds in both dry and flooded zones, and was therefore replaced with a more advanced set of moving speeds within the present version of the simulator (Sect. 2). Pedestrian agents were also featured with stability states to determine whether they can be mobile or immobile when they are inside the floodwater. The state of stability of the pedestrian agents was estimated based on two experimentally derived formulas (Xia et al. 2014). These formulas evaluate the incipient velocity limits ($U_c$) for toppling and sliding conditions of human subjects in the floodwater. The formulas are applied to each navigation agent by taking the body height and mass information of the pedestrian agent present at the navigation agent location, and also the states of floodwater variables at the coincident flood agent. The navigation agent then compares the incipient velocity to the magnitude of floodwater velocity ($V$). From this comparison, toppling and sliding stability states can be characterised for any pedestrian agent located at the navigation agent, as outlined in Table 2.

**Table 2:** The stability states of pedestrian agents in the flood-pedestrian simulator identified by comparing the incipient velocity limits ($U_c$) for toppling and sliding conditions to the velocity magnitude of floodwater (retrieved from Shirvani et al. (2020))

| Condition | Stability state of a pedestrian agent in floodwater |
|---|---|
| $V < U_c^{(Toppling)}$ and $V < U_c^{(Sliding)}$ | Stable condition - the pedestrian agent can have a moving speed |
| $V > U_c^{(Toppling)}$ and $V < U_c^{(Sliding)}$ | Toppling-only condition - the pedestrian agent is immobilised |
| $V < U_c^{(Toppling)}$ and $V > U_c^{(Sliding)}$ | Sliding-only condition - the pedestrian agent is immobilised |
| $V > U_c^{(Toppling)}$ and $V > U_c^{(Sliding)}$ | Toppling-and-sliding condition - the pedestrian agent is immobilised |

Note that the stability estimations in Table 2 were previously based on an empirical relationship for Body Mass Index (BMI) that only considers the body height of individual pedestrians (Shirvani et al., 2020). BMI ranges used in the present version of the simulator are based on age- and gender-related body mass characteristics (Sect. 2.2.1).

Finally, static-in-time destinations for the pedestrian agents were specified in the previous version of the simulator, such as entrance/exit doors taken as emergency exit during a flood evacuation in an indoor space (Shirvani et al., 2020). The pedestrian agents have therefore been fitted with new rules to enable the applicability of the simulator to study flood-induced pedestrian evacuation dynamics in an urban environment where there is no specific exit. These rules enable pedestrian agents to autonomously navigate to new pathways while the states of the floodwater change dynamically (Sect. 2.2.3).

**2.2 New characteristics and rules for pedestrian agents**

The pedestrian model was augmented to generate pedestrian agents with a random age and gender based on realistic datasets, that are also used to feature them with a set of randomised body mass (Sect. 2.2.1). More realistic ranges of moving speeds are also assigned to the pedestrian agents by considering their age and gender influences not only in dry zones but also in flooded zones (Sect. 2.2.2). New behavioural rules are also incorporated to enable pedestrian agents to alter their pathways and their choices for the emergency exit (Sec 2.2.3).



### 2.2.1 Age, gender, and body mass characterisation

Each pedestrian agent holds information of age, gender, and body mass at the time of its generation. To randomly assign an age, gender and body mass based on realistic distributions to each pedestrian, the UK national survey dataset (UK population by ethnicity, 2018) was used. As shown in Fig. 1, each pedestrian agent can have an age randomly selected from a range between 10 and 79 years old, and with a probability to keep the percentage of distribution of seven age groups. The excluded age groups, younger than 10 and older than 79 years old, make up 16 % of the UK population and represent children and elderly. To compensate for their exclusion, the percentage distribution of the other age groups was increased by around 2.3 %. Each pedestrian agent is also generated with a random 'male' or 'female' gender, each with equal chance of selection.

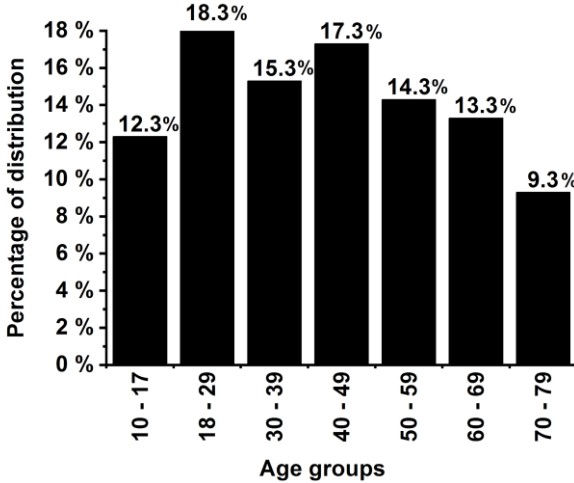

**Figure 1:** Age distribution assigned for the pedestrian agents in the flood-pedestrian simulator based on the UK's national survey (UK population by ethnicity, 2018)

Based on the age and gender of a pedestrian agent, its body mass, denoted by $m_p$ (kg), is evaluated using the following formula (Disabled World, 2019):

$$m_p = l_p^2 \, BMI, \tag{1}$$

where $l_p$ (m) stands for the body height of a pedestrian agent, which had already been incorporated within the previous version of the simulator (Shirvani et al., 2020). The BMI (kg / m$^2$) was randomly selected based on the ranges of age and gender listed in Table 3. For the age group between 10 and 17 years old, the BMI range was defined based on a standard for children (Prentice, 1998) and, based on samples of men and women who participated in the laboratory experiments reported in Bernardini et al. (2020) for the other age groups.



**Table 3:** Ranges of BMI used according to gender and age of individuals (details in Prentice (1998) and Bernardini et al. (2020))

| Age groups | Gender | BMI (kg / m²) |
|---|---|---|
| 10 to 17 | Both | Between 18.5 and 24.9 |
| 18 to 29<br>30 to 39<br>40 to 49 | Male | Between 18.21 and 32.10 |
| 50 to 59<br>60 to 69<br>70 to 79 | Female | Between 16.01 and 32.03 |


**2.2.2 Variable moving speeds**
Each pedestrian agent is enabled to evaluate their variable moving speed, age- and gender-related, based on two different sets
of rules depending on whether the agent is located in a dry zone or in a flooded zone. To enable a pedestrian agent discern
between dry zone and flooded zone, it resorts to the state of the floodwater's depth accessible from the navigation agent at its
specific location and time.
A pedestrian agent that identifies a zero depth of floodwater is automatically flagged to be in a dry zone. These
pedestrian agents are set to operate based on a 'dry-zone' moving speed rule under a walking condition. This rule assigns a
randomly selected walking speed to a pedestrian agent from a set of predefined ranges that are classified according to different
age and gender groups outlined in Table 4. The walking speed range of the 10-19 age group is defined according to the human's
average walking speed and is the same for both male and female (Mohler et al., 2007; Toor et al., 2011). For pedestrian agents
with 20 years of age and more, the ranges of their walking speed varies across different gender groups and are derived from
an empirically identified standard proposed by Bohannon and Andrews (2011). As people are expected to move faster under
evacuation conditions (Bernardini et al., 2020), pedestrian agents are applied an additional rule to increase their walking speed
based on the 'maximum excitement condition' identified in the experiments of Bernardini and Qualiarimi (2020). This
condition enables 'male' pedestrian agents to increase their walking speed by 60 % and 'female' agents to increase their
walking speed by 76 %. The experimental findings of Lee et al. (2019) also suggest a faster maximum excitement condition
for women, which may be associated with the fact that women have less tendency to be around floodwater compared to men
(Becker et al., 2015; Hamilton et al., 2020).

**Table 4:** Ranges of walking speeds for the pedestrian agents located in dry zones according to their age and gender (Mohler et al., 2007;
Toor et al., 2011; Bohannon and Andrews, 2011)

| Age range<br>(years) | Walking speed range (m/s) | |
|---|---|---|
| | **Female** | **Male** |
| 10 to 19 | 1.39 to 1.47 | 1.39 to 1.47 |





| 20 to 29 | 1.270 to 1.447 | 1.239 to 1.443 |
| 30 to 39 | 1.316 to 1.550 | 1.193 to 1.482 |
| 40 to 49 | 1.353 to 1.514 | 1.339 to 1.411 |
| 50 to 59 | 1.379 to 1.488 | 1.222 to 1.405 |
| 60 to 69 | 1.266 to 1.412 | 1.183 to 1.300 |
| 70 to 79 | 1.210 to 1.322 | 1.072 to 1.192 |


A pedestrian agent that identifies a non-zero depth of floodwater is automatically flagged to be in a flooded zone.
These pedestrian agents are set to operate upon a 'flooded-zone' moving speed rule under either 'walking' or 'running'
conditions. With this rule, each pedestrian is assigned a moving speed that is evaluated by an empirical formula extracted from
the experiments in Bernardini et al. (2020). Denoting the moving speed of each individual by $V_p$ (m/s), the formula reads
$$V_p = a.M^b,\qquad(2)$$
where $M$ is a function of specific force per width unit calculated by $M = \frac{V^2 h}{g} + \frac{h^2}{g}$ with $h$ and $V$ being the depth and the velocity
magnitude of floodwater respectively, $g$ is the gravitational constant and $a$ and $b$ are age-related parameters defining each of
the 'walking' and 'running' conditions, which are listed in Table 5. $M$ is estimated at the navigation agent, where the pedestrian
agent is present, from copies of $h$ and $V$ obtained from the flood agent at the coincident location.

**Table 5:** The values of age-related parameters, $a$ and $b$, identified by Bernardini et al. (2020) for evaluation of the moving speed of each
individual under 'walking' and 'running' conditions via Eq. (2)

| Age ranges (years) | Walking | | Running | |
| --- | --- | --- | --- | --- |
| | a | b | a | b |
| 5 to 12 | 0.82 | 0.18 | 0.41 | -0.21 |
| 13 to 20 | 0.54 | -0.07 | 0.81 | -0.19 |
| 21 to 28 | 0.36 | -0.13 | 0.48 | -0.19 |
| 29 to 36 | 0.35 | -0.19 | 0.53 | -0.23 |
| 37 to 44 | 0.43 | -0.13 | 0.62 | -0.20 |
| 45 to 52 | 0.57 | -0.03 | 0.61 | -0.17 |
| 53 to 60 | 0.32 | -0.17 | 0.62 | -0.20 |
| 61 to 68 | 0.16 | -0.43 | 0.61 | -0.17 |


The validity of Eq. (2) is limited to subjects under the age of 68 and only applicable to floodwater depths between 0.2 m to 0.7
m (Bernardini et al., 2020). In reality, floodwater depth can be outside these limits and it may happen that an elderly beyond
68 years of age is present in a flooded area. Therefore, extra rules were applied to extend the variety of moving speed of
pedestrian agents in flooded zones beyond the aforementioned age and floodwater depth limits for Eq. (2):
-    the moving speed of pedestrian agents with an age greater than 68 is evaluated by decreasing $V_p$ of the 61-68 age
group by 1.6 % per year, following the experimental findings of Dobbs et al. (1993),



-    pedestrian agents encountering a depth of floodwater shallower than 0.2 m are set to maintain dry-zone walking speed
rule as they are not expected to experience significant interference from the floodwater on their walking speed (Lee
et al., 2019), and

-    pedestrian agents encountering floodwater greater than 0.7 m are given a moving speed informed by the stability
limits reported in the UK's Flood Risks to People method (Environment Agency, 2006). Namely, these pedestrian
agents are only set to have a moving speed when velocity magnitude $V$ is less than 1.5 m/s, or otherwise, they remain
immobile.

### 2.2.3 Autonomous change of direction condition

Each pedestrian agent is also featured with two extra rules to enable it to autonomously navigate into new pathways while
moving within a flooded zone, where it encounters a non-zero floodwater depth from the navigation agent at its specific time
and location. The first rule makes a pedestrian agent detect and choose another destination if the floodwater depth along its
way becomes higher than a threshold of a floodwater depth to body height. The choice for the threshold is case-dependent and
exploring many thresholds may be necessary (Sect. 3.3.2) as personal flood risk perception is uncertain and this affects the
modelling of decisions, of when and where to enter the floodwater or make a move into another destination (Becker et al.,
2015; Netzel et al., 2021). Applying this rule enables the pedestrian agents to make decisions on which pathway to take within
an environment layout where there is no specific emergency exit at time of evacuation. The second rule applies to those
pedestrian agents which remain undecided about selecting a pathway after a period of time (user-selected, Sect. 3.3.2). Such
pedestrian agents are then set to detect the most popular destination chosen by the pedestrian agents within its surrounding.
This rule is applied on the basis that group decisions have significant influence on the path finding decision of an individual
in and around the floodwater (Becker et al., 2015; Lin et al., 2020).

### 2.3 Evaluation of the new characteristics and rules

The new characteristics and rules for pedestrian agents in the present version of the simulator were evaluated with a focus to
assess their relevance for the analysis of pedestrian evacuation dynamics during a flood emergency (Sect. 2.3.1). Direct
validation of agent-based models is a grand challenge as such models are aimed to study non-observable scenarios, where
there is uncertainties associated with the emergent nature of behaviours and where validation datasets of such type are not
available (An et al., 2020; Zhuo et al., 2020; Aerts 2020). One alternative approach is a component-wise validation (Bert et
al., 2014). This approach was used at the development stage of the dynamically coupled hydrodynamic and pedestrian models
within the simulator (Shirvani et al., 2021).

To validate the relevance of in-model behavioural rules, one suitable strategy is to Take A Previous Model and Add
Something (TAPAS) (Polhill et al., 2010; Abebe et al. 2020). This strategy was previously applied by systematically increasing
the level of sophistication of agent characteristics and rules, and running the simulator progressively to identify their relevance
by analysing the respective changes to the simulation outcomes (Shirvani et al., 2020). The TAPAS approach is also applied





here to evaluate the new characteristics and rules added to the present version of the simulator, by setting it up and running it
for the same test case used in Shirvani et al. (2020), Sect. 2.3.1, for five different configuration modes that are summarised in
Table 6.

**Table 6:** Configuration modes used to set up and run the simulator to evaluate the newly added characteristics and rules

| Modes | Pedestrian behavioural rules | | | | |
|-------|------------------------------|---|---|---|---|
|  | Two-way interaction | Moving speed in dry zones | | Moving speed in flooded zone (Eq. (2)) | |
|  |  | Walking condition | Maximum excitement condition | Walking condition | Running condition |
| Mode 0 | Disabled | Constant | Disabled | Age independent | Not applicable |
| Mode 1 | Disabled | Age- and gender-related (Table 4) | Enabled (Sect. 2.2.2) | Age-related | Not applicable |
| Mode 2 | Enabled | | | | |
| Mode 3 | Disabled | | | Not applicable | Age-related |
| Mode 4 | Enabled | | | | |


The baseline Mode 0 retrieves the previous set up of the simulator (Shirvani et al., 2020), where pedestrian agents are

only assigned a constant walking speed in dry zones and without the 'maximum excitement condition' (Sect. 2.2.2). In this
mode, the pedestrian agents in flooded zones pick up a set of variable moving speeds under a 'walking' condition that is not
age dependent. In addition to these, the 'two-way interaction' condition is also disabled (Sect. 2.1.1). Commonly in Mode 1 to
4, a random age- and gender-related walking speed is assigned to the pedestrian agents in dry zones (from Table 4). In Mode
1, the 'walking' condition (Sect. 2.2.2) is enabled, while Mode 2 allows to also take into account the 'two-way interaction'
condition (Sect. 2.1.1). With Mode 3 and Mode 4, the 'running' condition (Sect. 2.2.2) is enabled instead, but only Mode 4
also enables the 'two-way interaction' condition (Sect. 2.1.1). The 'autonomous change of direction' condition is disabled with
the simulation modes as the test case used in Shirvani et al. (2020) involves an evacuation of pedestrians in an indoor space
where there is one specific emergency exit (Sect. 2.3.1). The latter condition will only be enabled for the real case study
reported in Sect. 3 that involves an evacuation in an outdoor streetscape with many emergency exits.
**2.3.1 Overview of the flood evacuation in a shopping centre test case**
This test case was explored with the previous version of the flood-pedestrian simulator (Shirvani et al., 2020, Shirvani et al.,
2021). It is reconsidered to assess the relevance of the new characteristics and rules added to the pedestrian agents within the
present version of the simulator.

The test case considers a $332 \times 332$ m$^2$ hypothetical shopping centre that includes stores along its west and east sides,

corridors and seven entrance/exit doors to the open space area (Fig. 2). The total walkable area of the shopping centre, including
the open area and the corridors, is equal to 70,350.8 m$^2$. The pedestrian model within the simulator was set to have a constant





rate of 10 pedestrian agents entering or leaving from each of seven doors with an equal probability of one in seven. This set
up maintains a constant number of 1,000 pedestrians wandering in the walkable area, when there is no floodwater. This number
was selected to give an area of around $8.4 \times 8.4$ m$^2$ for each person using a calculator toolbox of the average space required
for individuals in malls (Engineering ToolBox, 2003). The floodwater propagation was assumed to breach from the southern
side along a 100 m opening (Fig. 2). When the floodwater starts to propagate, no more pedestrian agents are generated in the
walkable area and the remaining ones are set to move to the emergency exit located at the northern side (Fig. 2), which was
the only door open during the evacuation process.

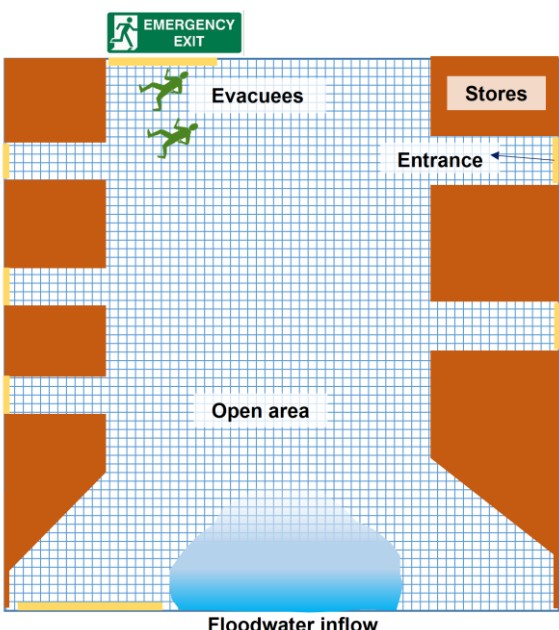


**Figure 2:** Sketch of the shopping centre: the meshed area in blue indicates the open area where pedestrians can walk to the entrance doors (coloured in yellow). Once the flood starts, evacuating pedestrians will go to the emergency exit (on the north side). The blocks in brown indicate terrain features assuming they are stores and the blue-shaded area in the southern part of the figure shows the location where the floodwater started to propagate.

The flooding inflow was generated based on an inflow hydrograph of a discharge, $Q$ (m$^3$/s) propagating over a

duration of 7.5 min, and peaking to 160 m$^3$/s at 3.75 min (Fig. 3). The hydrograph was produced based on the Norwich
inundation case study, and because it results in a range for the HR that is inclusive of all the ranges listed in Table 1, i.e. HR
< 7 (Shirvani et al., 2021). Deploying a hydrograph with shorter duration or a bigger peak would lead to significantly bigger
HR, which is indicative of loss of life where a person does not have a chance to take an action and would immediately have a
stability state under the sliding-only condition (hence is outside the scope of this study). When the floodwater starts to
propagate over the walkable area, simulation time ($t$) of 0 min, the pedestrian agents start the evacuation and the simulation
terminates when all the pedestrian agents have evacuated the walkable area.





The simulator was executed at a resolution of 2.59 m × 2.59 m for each of the grids of navigation and flood agents.
The time-step was taken to be the minimum between the adaptive time-step of the hydrodynamic model and the 1.0 s time-
step of the pedestrian model (a visualisation of the simulation can be found in the video supplement in Shirvani (2021)). In
each run, the simulator is set to record the information stored in the flood agents and the pedestrian agents at each time-step.
Recorded outputs from a simulation run include the positions of the pedestrian agents, their flood risk states (HR-related) and
or their stability states (with toppling-only condition and toppling-and-sliding conditions). Four runs were conducted based on
the configuration Mode 1 to Mode 4 (Table 6), and their recorded outputs are analysed next with respect to the baseline outputs
obtained from the run of the simulator with configuration Mode 0.

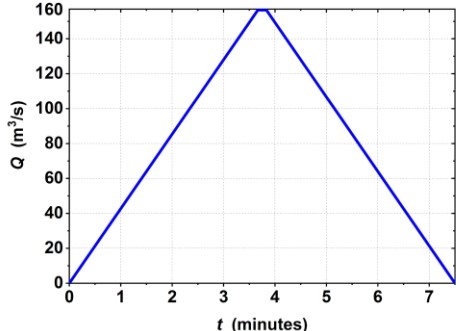


**Figure 3:** Inflow hydrograph that is used to generate the floodwater propagation from the southern side of the shopping centre
**2.3.2 Analysis of flood risks to people**
Figure 4a shows the number of evacuating pedestrians with a low flood risk state (HR < 0.75) predicted by the simulator using
all the configuration modes (Table 6). Figure 4a-left includes the evacuation patterns predicted by enabling the walking
condition for the age-related moving speeds (Mode 1) versus those predicted by further enabling the two-way interaction
condition (Mode 2). In Mode 1, the evacuation patterns are in good agreement with the baseline predictions (Mode 0, with
non-age related moving speeds) at flooding times when there are less than 50 pedestrians in the surrounding with a low flood
risk state, during 3.5 min to 6 min. Differences among the predictions start to occur when more than 150 pedestrians were
present, at 2 min and 8 min. This difference seems to impact the overall prediction, leading to an evacuation time that is 6 min
longer and a higher number of pedestrians being predicted to be under this flood risk state, during 8 min to 20 min. In Mode
2, compared to Mode 1, the number of evacuating pedestrians is seen to reduce further at flooding times involving more than
150 pedestrians, at 2 min and 10 min. This is expected as the more the crowding of pedestrians the more the energy loss in the
floodwater dynamics for low risk floodwaters, which in turn enables the pedestrians located behind to take a faster moving
speed. Figure 4a-right includes the evacuation patterns predicted by activating the running condition for the age-related moving
speeds (Mode 3), and those predicted by also enabling the two-way interaction condition (Mode 4). In Mode 3 and Mode 4,
the evacuation time decreased significantly (by almost 67 %) leading to evacuation patterns that are close to the baseline



predictions (Mode 0). With Mode 3, discrepancies only occur after 2 min of flooding and after 8 min when there are more than
200 pedestrians moving based on the running condition. In Mode 4, the evacuation patterns remain close to those predicted
under Mode 3, except at flooding times involving more than 200 pedestrians, at 2 min and 8 min. Overall, using the age-related
moving speed with the walking condition and without or with the two-way interactions condition (Mode 1 or Mode 2) predicts
longer evacuation patterns for pedestrians in a low flood risk state compared to the other three modes that lead to close patterns
(Mode 0, Mode 3 and Mode 4). This condition seems useful to use with the simulator to study mass evacuation planning
scenarios under low risk flooding. With the two-way interaction condition, slightly accelerated evacuation patterns should be
expected at the times of the flooding when crowding occurs for pedestrians with a low flood risk state.
Figure 4b shows the number of evacuating pedestrians under a medium flood risk state ($0.75 < HR < 1.5$). With Mode
1, compared to Mode 0, a faster evacuation pattern is observed until 6 min before the number of pedestrians becomes 300.
This suggests that before crowding occurs, pedestrians with a medium flood risk state are predicted to evacuate faster with
Mode 1. However, when the number of pedestrians becomes more than 300, during 6 min to 8 min, Mode 1 leads to a slightly
slower evacuation as observed previously with the pedestrians under a low flood risk state (compare Fig. 4a-left and Fig. 4b-
left). This difference is not observed for the evacuation patterns predicted by the simulator with Mode 2, which leads to a
number of pedestrians that does not exceed those observed with Mode 0. By using instead the age-related running condition
under Mode 3 (Fig. 4b-right), no major changes can be observed in the evacuation patterns for the pedestrians with a medium
flood risk state (compared to with Mode 0). Further enabling the two-way interaction condition (Mode 4) induces a slight
reduction in the predicted number of pedestrians during the flooding time when the crowding is at its peak (Fig. 4b-right). As
for the evacuation speed, it is predicted to be the same with all the modes for the pedestrians under a medium flood risk state.
However, Mode 0 should be avoided as it can lead to the simulator predicting less pedestrians in this risk category, in particular
at early flooding times before crowding occurs (compare the evacuation patterns in Fig. 4b during 2 min to 6 min). Hence,
running the simulator with any of the configuration Mode 1 to Mode 4 does not seem to yield a major difference in the
evacuation patterns for pedestrians with a medium flood risk state.
Figure 4c shows the number of evacuating pedestrians predicted to be under a high flood risk state ($1.5 < HR < 2.5$).
For pedestrians with this risk category, running the simulator with any of the Mode 1 to Mode 4 lead to major differences in
the evacuation patterns compared to those predicted under Mode 0. With Mode 1 to Mode 4, only a handful of pedestrians are
predicted to have a high flood risk state, during 3 min to 5 min of the flooding, in contrast to what the simulator's prediction
with Mode 0 suggests: up to 140 pedestrians within a time window of 4 min. Hence, using the age-related moving speed, under
either the walking condition or the running condition, seems to make a difference in the predicted evacuation patterns of
pedestrians with a high flood risk state. The impact of the two-way interaction condition on the evacuation pattern of such
pedestrians can be detected by analysing the difference between the predictions made under Mode 1 vs. Mode 2, for the age-
related walking condition (Fig. 4c-left), and between Mode 3 vs. Mode 4, for the age-related running condition (Fig. 4c-right).
As can be seen, only a slightly higher number of pedestrians with a high flood risk state are predicted when the two-way
interaction condition is also enabled, suggesting that it would not lead to a major difference in this case.



Figure 4d shows the number of evacuating pedestrians predicted to have a highest flood risk state (2.5 < HR < 20).
In this case, with any of the Mode 1 to Mode 4, the simulator predicts these pedestrians to be at earlier times in the flooding
compared to Mode 0. This implies that using the age-related moving speeds can provide a better prediction of the timing when
the evacuating pedestrians would be at the highest flood risk state. The evacuation patterns predicted by the simulator using
Mode 1 and Mode 3 are similar, indicating any of the running or walking conditions would lead to similar outcomes to inform
on evacuating pedestrians with a highest flood risk state. However, the walking condition combined with the two-way
interaction condition (Mode 2) leads to slightly higher number of pedestrians with this flood risk state, as these pedestrians
would be more affected by the local changes induced the floodwater dynamics from those pedestrians, with a low risk state,
crowding ahead (Shirvani et al., 2021). The running condition combined with the two-way interaction condition (Mode 4) does
not lead to major changes in the evacuation patterns compared to the predictions with Mode 2. Hence, Mode 2 seems to be a
sensible choice to use with the simulator to plan evacuation case studies involving severe flooding scenarios too.

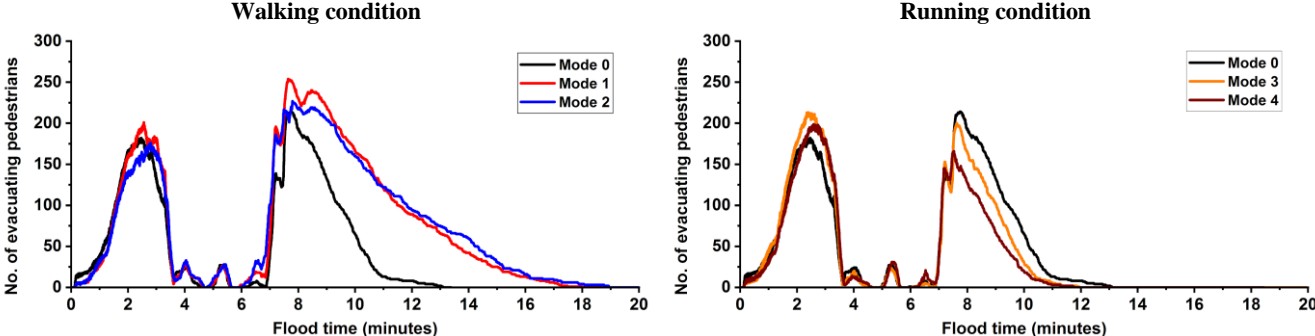

(a) Evacuating pedestrians with a low flood risk state (HR < 0.75)

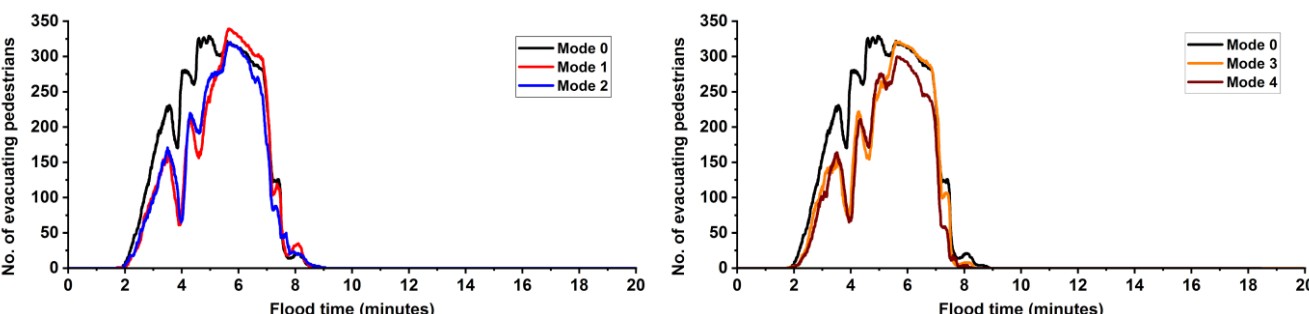

(b) Evacuating pedestrians with a medium flood risk state (0.75 < HR <1.5)
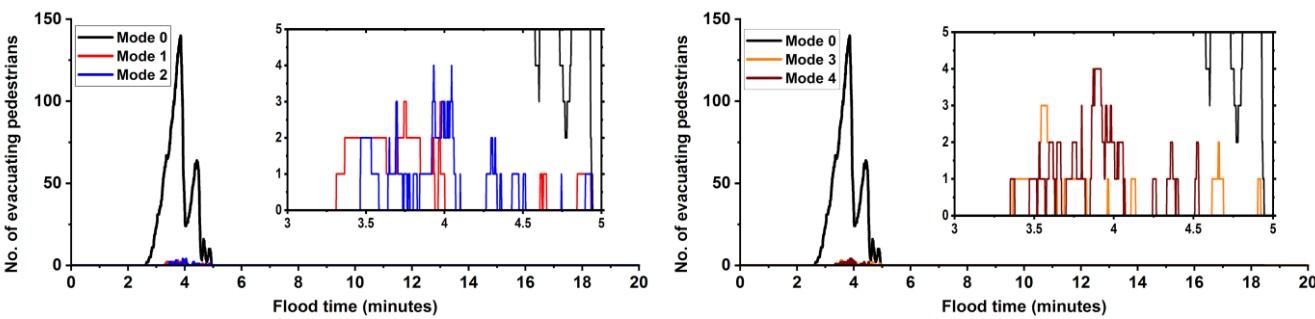

**(c)** Evacuating pedestrians with a high flood risk state (1.5 < HR < 2.5)

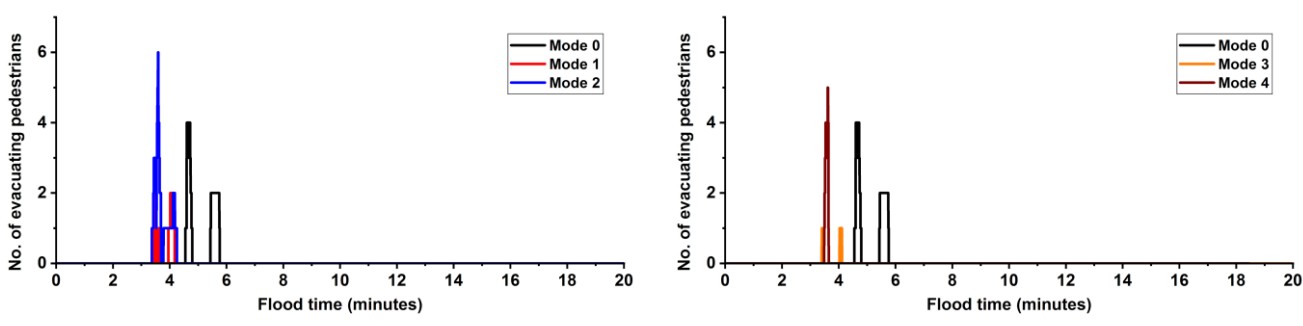

**(d)** Evacuating pedestrians with a highest flood risk state (2.5 < HR < 20)

**Figure 4:** Number of evacuating pedestrians predicted by the simulator under Mode 0 (baseline outcomes from the previous version of the simulator, Shirvani et al., 2020); Mode 1 or Mode 2 (age-related walking condition for the moving speeds without or with the two-way interaction condition); and Mode 3 or Mode 4 (age-related running condition for the moving speeds without/with the two-way interaction condition). Analysis is presented in sub-figures (a)-(d), each considering a different flood risk state.

Figure 5 contains the stability states of the pedestrians, plotted in terms of the number of evacuating pedestrians with a toppling-only condition (Fig. 5a) and with a toppling-and-sliding condition (Fig. 5b). The stability states are shown for the simulator predictions obtained by enabling the walking condition without or with the two-way interaction condition (Mode 1 or Mode 2), and by enabling the running condition without or with the two-way interaction condition (Mode 3 or Mode 4). The baseline predictions for the pedestrians' stability states obtained under Mode 0 are also included. With Mode 1 and Mode 3, the number of pedestrians predicted to be with a toppling-only condition and with a toppling-and-sliding condition are very similar to the baseline predictions. The pedestrians with such stability states were detected during 2 min to 8 min for the toppling-only condition, and during 2 min to 6 min for the toppling-and-sliding condition. As explored in Fig. 4, evacuating pedestrians with a low to medium flood risk state could be detected during 2 min to 8 min. Hence, using the simulator with the age-related moving speeds (Mode 1 or Mode 3) lead to similar information on the stability states even when the pedestrians have a low to medium flood risk state. With the two-way interaction condition (Mode 2 or Mode 4), less pedestrians were predicted to be with toppling-only and toppling-and-sliding conditions, namely when crowding occurs after 4.6 min, where there is a notable reduction in the number of pedestrians with a toppling-and-sliding condition (see also Fig. 6). Hence, the



activation of the two-way interaction condition predicts a lower number of pedestrians immobilised by the floodwater, namely
when the walkable area involves localised crowding of pedestrians. This can also be observed in Fig. 6 that contrasts 2D spatial
distributions of the evacuating pedestrians obtained with and without the two-way interaction condition (Mode 1 vs. Mode 2
only, as the distributions carried out with Mode 3 vs. Mode 4 led to similar observations).

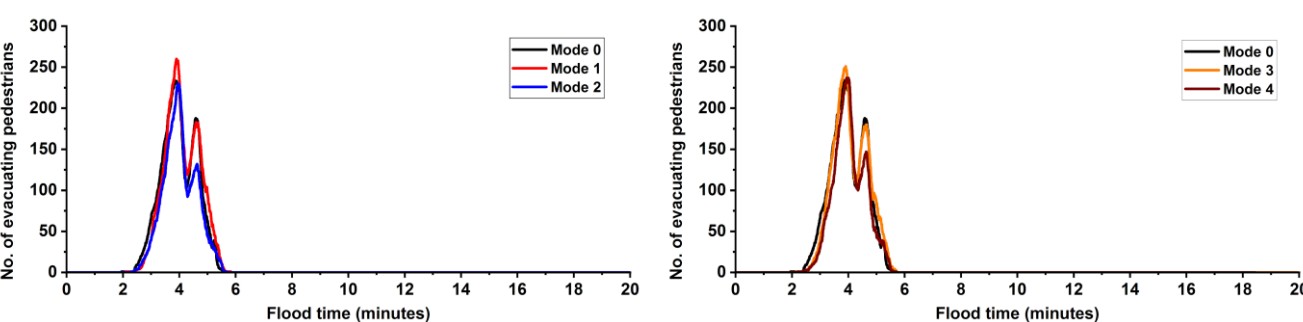

**(a)** Evacuating pedestrians with a toppling-only condition

**(b)** Evacuating pedestrians with a toppling-and-sliding condition

**Figure 5:** Number of evacuating pedestrians with stability state predicted by the simulator under Mode 0 (baseline outcomes from the
previous version of the simulator (Shirvani et al., 2020); Mode 1 or Mode 2 (age-related walking condition for the moving speeds without
or with the two-way interaction condition); and Mode 3 or Mode 4 (age-related running condition for the moving speeds without or with the
two-way interaction condition). Sub-figures (a) and (b) include the stability states with a toppling-only condition and a toppling-and-sliding
condition, respectively, when immobilised in floodwater
Figure 6 overlaps the spatial distributions of the evacuating pedestrians and the flood HR map at two critical times of
the flooding, 3.6 min and 4.6 min. The circled areas indicate higher-than-medium risk zones of floodwater flow with HR >
1.5. As shown in Fig. 6a, at 3.6 min, the two-way interaction condition magnifies the higher-than medium risk zones at the tail
of the crowding leading to more pedestrians at risk of being in a toppling-only or a toppling-and-sliding conditions. Whereas,
at 4.6 min (Fig. 6b), the two-way interaction condition led to having less pedestrians with a toppling-and-sliding condition in
the heart of the crowded area, where their stability state changed to a toppling-and-sliding condition.


In terms of total time to evacuate the 1,000 pedestrians, the simulator predicted around 13 min with Mode 0, 18 min
with Mode 1, 19 min with Mode 2, 11.7 min with Mode 3 and 11.5 min with Mode 4. Contrasting the predicted times reinforces
previous findings from Fig. 4: compared to Mode 0, the age-related walking speeds (Mode 1) predicts a slower evacuation
time and enabling it with the two-way condition (Mode 2) increases slightly the evacuation time as crowding is more likely
under the walking condition. The age-related running speeds (Mode 3) leads to a shorter evacuation time than with the other
modes, that are slightly reduced when enabling the two-way condition (Mode 4). Next, the simulator is only run under Mode
2 to analyse a mass evacuation of pedestrians that mostly have low to medium flood risk states (Sect. 3).

● Stable condition      ● Toppling-only condition      ● Toppling-and-sliding condition

**(a)** At 3.6 min after flooding

**(b)** At 4.6 min after flooding

**Figure 6:** Spatial distribution of pedestrian agents, represented by coloured dots, predicted by the simulator under Mode 1 (left panel) and
Mode 2 (right panel) at two different flood times: (a) 3.6 min after flooding and (b) 4.6 min after flooding. The grey colour represents the


floodwater extent based on the flood HR quantity and the red-circled areas show where the HR is greater than 1.5, indicative of a high-to-
highest flood risk.

**3 Real-world case study**

**3.1 Background and scenario description**

The case study consists of a site located outside of the main entrance of Hillsborough football stadium in Sheffield. The location
of the site is framed with a red square in Fig. 7, including an area of 16,384 m$^2$ that is adjacent to the eastern side of the stadium,
where the main entrances/exits are located (yellow line in Fig. 7). The stadium entrances are opened to a T-junction streetscape
that constitutes the walkable area including all the main roads and pedestrian pathways (this area is framed in Fig. 7). The
main roads are open from the south, east and north sides of the walkable area (shown with the green lines), and these openings
are likely to constitute the most popular destinations where spectators would go before and after a football match.

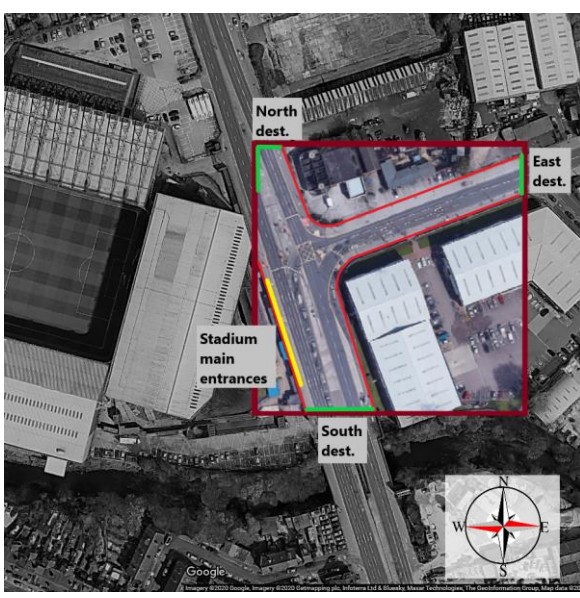


**Figure 7:** The study site (red square) including the walkable area (red area within the red square) where people normally use to go to their
different destinations located in the south, east and north sides of the walkable area (green lines) after they leave the stadium from the main
entrances (yellow line), © Google

The stadium can accommodate up to 39,732 spectators, and usually has an average attendance rate of around 24,000
per each home football match in normal weather conditions (Football Web Pages: Sheffield Wednesday, available at:
www.footballwebpages.co.uk/sheffield-wednesday). This site would therefore be occupied by large numbers of pedestrians,
including football fans before and after a match, even in the aftermath of a flood as, for example, happened for the November
2019 Sheffield floods (Pugh, 2019). This site also experienced flooding in the 2007 UK floods, due to extreme rainfall, of
around 100 mm in one day (Environment Agency, 2007). The historical 2007 flood suggests that rainfall runoff would cause
floodwater inundation to spread along the northern side of the stadium, and to accumulate within the selected site where it



submerged pedestrian pathways, parking lots and the stadium pitch. Worries of a similar event were expressed during the 2019
Sheffield floods that were caused by seven-day continuous rainfall of 63.8 mm. This led to a cancelation of a football match
as water level slightly overtopped the flood defence protecting the stadium from the River Don. The event, if happened during
the football match, could put many in and around the stadium at a high risk.
Being both adjacent to River Don and located down the hills where rainwater runoff accumulate, this site has been
flagged to be prone to future pluvial or fluvial flooding types according to EA's online flood maps for planning (available
online at: https://flood-warning-information.service.gov.uk/long-term-flood-risk). Figure 8 shows a screenshot of the EA's
floodwater depth map for the walkable area covering the cases of 'low', 'medium' and 'high' annual flooding probabilities.
Even though with the 'low' flooding probability scenario, floodwater inundation would still reach a depth of 0.9 m, which is
high enough to force people to select unusual pathways or temporarily change their direction towards another destination to
avoid the floodwater as they evacuate.

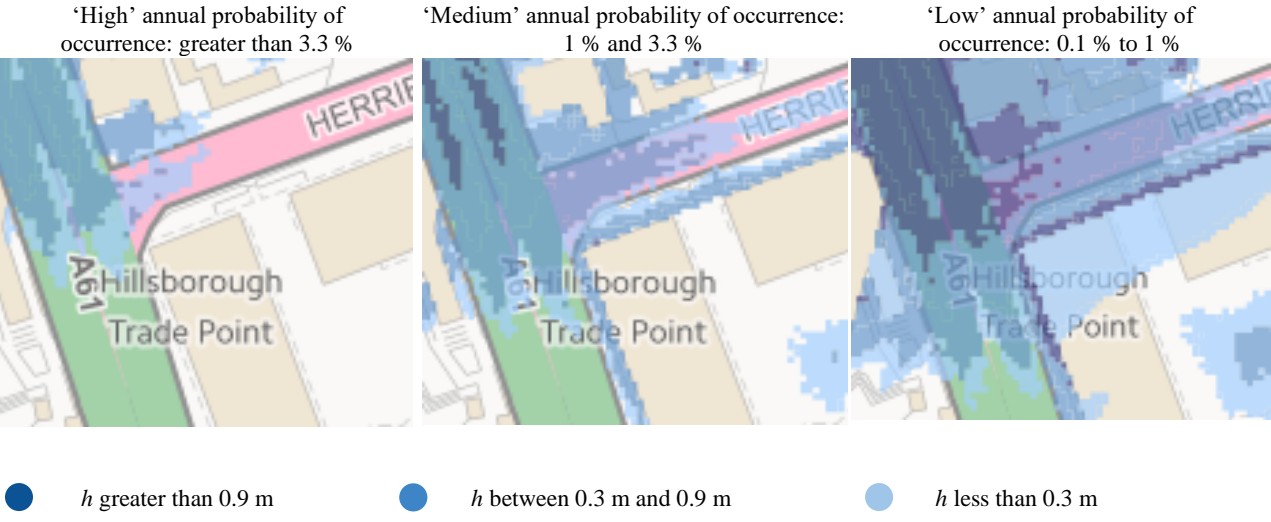

**Figure 8:** Screenshots showing the floodwater depth map for the walkable area covering the cases of 'low', 'medium' and 'high' annual
flooding probabilities. These screenshots were retrieved from https://flood-warning-information.service.gov.uk/long-term-flood-risk (credit:
© Crown and database rights under Open Government Licence v3.0).
In this study, it is assumed that the site in Fig. 7 is hit by a flood, similar to the one that had happened in 2019, during a football
match where the spectators are caught unaware of the rainfall accumulation around the stadium. The floodwater occurs from
the north-east side and moves and accumulates downhill towards the main entrance of the stadium. Once the floodwater
approaches the main entrances, an emergency evacuation alarm is issued urging people to evacuate the stadium immediately.
The spectators are then put into queues inside the stadium, waiting to be evacuated towards the walkable area. The evacuated
spectators gradually enter the walkable area where they get in direct contact with the propagating floodwater along their ways
to any of the south, east or north destinations. The walkable area was measured to have a size of 8,161 m², and a reduced





number of 4,080 spectators (2 person per square metre) are assumed to have attended the football match, given the extreme
weather conditions. This population size respects the highest safety limit for moving and queueing states at sports grounds
(Still, 2019). Using a bigger population size would lead to extreme pedestrian congestion that impacts the movements of
individual pedestrians, governed by the social force model (Minegishi and Takeichi, 2018). Hence, limiting the number to
4,080 pedestrians is also helpful to more realistically evaluate the net effect of pedestrian behaviours in relation to the changes
in the state of floodwater variables during the evacuation process.

The flood-pedestrian simulator is applied to simulate this scenario for analysing the pedestrian evacuation patterns

including their preference for the destination during flood evacuation, by activating the 'autonomous change of direction'
condition (Sect. 2.2.3).

**3.2 Simulator configuration**

To simulate the evacuation scenario, the simulator was gradually configured; first, using its hydrodynamic model alone to
ensure that it replicates the floodwater hydrodynamic upon a realistic terrain and driven by a hydrograph representing the
extreme rainfall event that occurred in November 2019 (Sect. 3.2.1); and, then, by further configuring its pedestrian model
with scenario-specific characteristics for the pedestrian agents considering population age, gender, height, probability of
destination selection, and thresholds of floodwater depth to body height required for the 'autonomous change of direction'
condition (Sect. 3.2.2).

**3.2.1 Hydrodynamic model set-up**

The hydrodynamic model was set up to run on a grid of $128 \times 128$ flood agents. The grid of flood agents (equally for the grid
of navigation agents) was set to store the terrain features of the study site, loaded from a digital elevation model (DEM) at 1
m resolution, which is available online from the UK's Department for Environment Food & Rural Affairs (DEFRA) LiDAR
Survey (available at: https://environment.data.gov.uk).

To generate the flood propagation from the north-east side of the site, an inflow hydrograph (Fig. 9) was generated

based on the November 2019's rainfall data. The hydrograph was set to replicate a total runoff volume accumulation of 1,045.3
m$^3$ based on a 0.0638 m rainfall over the entire 16,384 m$^2$ site. This volume was estimated using the direct runoff method:
*rainfall volume* (m$^3$) = *rainfall height* (m) $\times$ *area* (m$^2$). The hydrograph was generated as:
$$Q_t = Q_{initial} + (Q_{peak} - Q_{initial})(\frac{t}{t_{peak}}.exp(\frac{1-t}{t_{peak}}))^\beta, \tag{3}$$
where $Q_t$ (m$^3$/s) is the inflow discharge propagating along the north-east boundary intersecting with the road; $Q_{peak}$ (m$^2$/s) =
0.29 is the peak discharge, that was calculated by distributing the runoff volume (1,045.3 m$^3$) per second over an hour of
flooding; $Q_{initial}$ (m$^3$/s) is the initial discharge, taken 0 m$^3$/s; $t$ (min) is the simulation time varying between 0 to 10 min; $\beta = 10$
is a constant to soften the shape of the hydrograph and $t_{peak}$ (min) = 5 is the time of peak discharge. This choice, for $t_{peak}$,



Natural Hazards
and Earth System
considers the peak discharge has been reached halfway during the flooding to cause the propagating floodwater to reach to the
main stadium's entrances by 10 min leading to triggering the evacuation alarm.

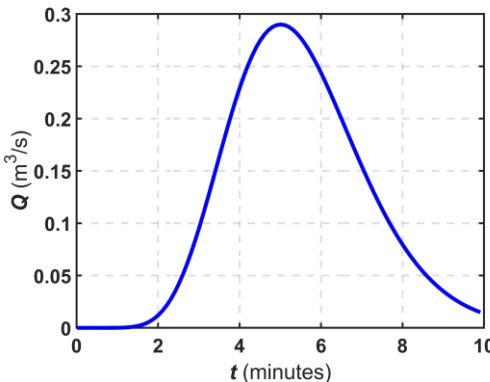


**Figure 9:** Inflow hydrograph produced by Eq. (3) used to generate the floodwater propagation occurring from the north-east side of the site

To evaluate whether the floodwater resulting from the hydrograph would resemble reality, the simulator was run,

without pedestrian consideration, to analyse the extent to which the predicted range for the floodwater depth (Fig. 9) agrees
with that predicted by in the EA's flood risk maps (Fig. 8). At the end of simulation ($t = 10$ min), the states of floodwater
variables stored by all the flood agents were extracted. The information relevant to the depth of floodwater was used to plot a
flood map that is shown in Fig. 10a. The flood map in Fig. 10a suggests a maximum water level of 0.9 m over the walkable
area and that the highest and lowest risk zones are located in the north-west and south, respectively, thus in good agreement
with the risk extents and range of floodwater depths reported by the EA's (Fig. 8). The information of the states of the
floodwater variables were also analysed for the HR flood map to gain an insight into the potential flood risk states that
pedestrians would be having within the walkable area. Fig. 10b shows the HR flood map at 10 min, all over which the HR
reaches a maximum value that is slightly greater than 1.5, suggesting that the majority of pedestrians would be having low to
medium flood risk states at the onset of evacuation. Overall, the generated inflow hydrograph is able to replicate realistic
floodwater depths and extents within the walkable area where the majority of pedestrians would be expected to mostly be at a
low to medium flood risk state. Next, the simulator is applied with pedestrian considerations while re-running the
hydrodynamic model with a lag of 10 min in the simulation time, $t = -10$ min, to account for the fact that the evacuation started
at $t = 0$ min (so enough time had passed for the floodwater to partially submerge the highest risk zones in the site).

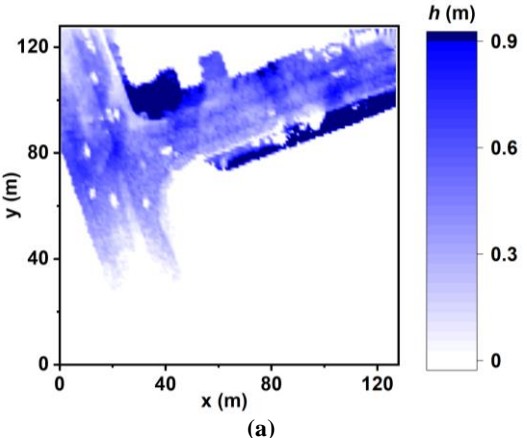
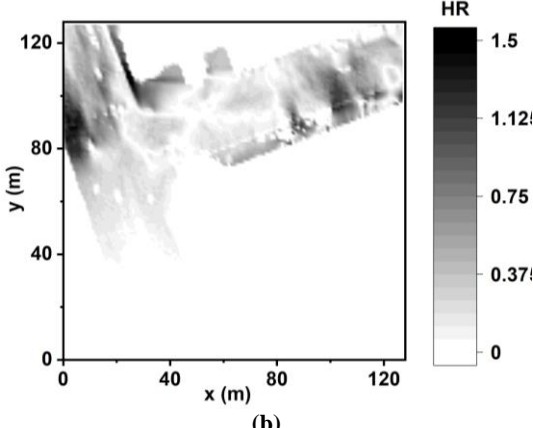

(a)                                                    (b)

**Figure 10:** Flood maps generated after 10 min of a single hydrodynamic without pedestrian consideration run based on: (a) floodwater depth and (b) HR extent over the entire site

### 3.2.2 Pedestrian model set-up

The pedestrian model was also set up for a grid of $128 \times 128$ navigation agents encoding the topographic features of the site, i.e. the boundaries, location of exits/entrances, and obstacles, into the navigation map. Hence, the navigation map has also a resolution of 1 m over which the pedestrian agents receive information about their destinations. The pedestrian model was set to gradually generate 4,080 pedestrian agents with a rate of 4 pedestrian agents per second starting at simulation time $t = 0$ min, to ensure safe evacuation timing for crowd evacuation planning in stadiums (Minegishi and Takeichi, 2018). Once a pedestrian agent is generated, it is assigned a random (initial) destination, to which they start to move to, between the three available destinations with an equal probability (south, east or north shown in Fig. 7). Along their ways, each pedestrian agent receives information on the state of floodwater variables, via the navigation agent at their specific time and location, based on which it makes a decision on whether to alter their direction to another destination.

As the case study consists of an outdoor urban environment with no specific destination, the pedestrian agents are set to dynamically alter their destination by activating the 'autonomous change of direction' condition (Sect. 2.2.3). This condition allows pedestrian agents to auto-select new pathways after processing the received information on the state of the floodwater variables. As explained in Sect. 2.2.3, this condition requires specifying a threshold of floodwater depth to body height beyond which a pedestrian agent considers shifting their walking direction and looking for a new destination within 100 seconds. After this period, if the pedestrian agent remains undecided, it is set to pick the destination selected by the majority of its neighbouring pedestrian agents, on the basis that it was influenced by the choice of others around (Sect. 2.2.3).

As people may underestimate the floodwater risk and can enter floodwater from different risk perceptions (Becker et al., 2015; Netzel et al., 2021), three thresholds of floodwater depth to body height were investigated to accommodate this uncertainty (Fig. 11). The '20 % threshold' was defined to represent people with high-risk perception, who may not enter floodwater with a depth that is more than 20 % of their body height. This threshold, for this case study, is estimated based on





the ratio of the dominant minimum value for the depth of floodwater that can occur over the walkable area (0.3 m) to the height
of the shortest pedestrian agent available (1.4 m). With this threshold, the likelihood of the entire population to be in a condition
to change their direction is ensured. The '40 % threshold' was defined to represent people with low-risk perception, who may
enter a floodwater with a depth that is even more than 40 % of their body height. This threshold, for this case study, is estimated
based on the ratio of the dominant maximum depth of floodwater (0.9 m) to the height of the tallest population of pedestrian
agents available (2.1 m). This threshold enables the entire population to have the freedom to keep moving even within the
deepest floodwater in the walkable area (0.9 m). The '30 % threshold' was defined based on an average-risk perception, and
is selected as it represents the knee height of humans, irrespective of their body height (Teichtahl et al., 2012).

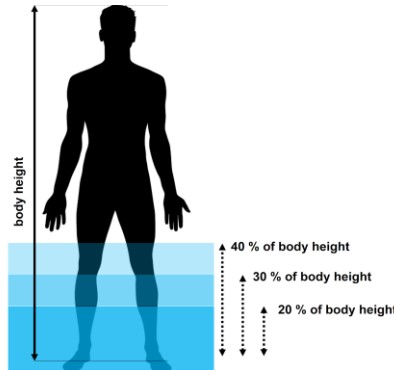


**Figure 11:** Thresholds of floodwater depth to body height that are specified for pedestrian agent to accommodate uncertainty associated
with different risk perception of people in the real-world case study

The characteristics of pedestrian agents were adapted to consider the age, gender and height distribution of football

fans in the UK. Therefore, the randomised age distribution reported in Sect. 2.2.1 was increased by 5 %, 8 % and 4 % for the
age groups of 30 to 39, 40 to 49, and 50 to 59 in order to replicate the higher attendance of these age groups to live sports
events in England (Lange, 2020). Also, the randomised gender distribution was changed to 67 % males and 33 % females
based on a survey on the gender distribution of football fans in the UK (Statista Research Department, 2016). In terms of body
height, the pedestrian agents were based on the same UK body height distribution used previously (Shirvani et al., 2020).
**3.2.3 Simulation runs**
The simulator was run under configuration Mode 2, which was deemed suited for the scope of this case study (Sect. 2.3.2).
Three simulations were performed by considering each of the 20 %, 30 % and 40 % thresholds in the 'autonomous change of
direction' condition, respectively (visualisation of the simulations can be found in the video supplement in Shirvani (2021)).
Outputs from each simulation run included spatial and temporal information, at each time step, about the pedestrian agents as
they evacuate ($t > 0$ min). The predicted outputs include the position, flood risk state (HR-related), stability state (with a
toppling-only condition and a toppling-and-sliding condition), and the choice for the destination selected by the pedestrian
agents during the evacuation process. These outputs are analysed for each the 20 %, 30 % and 40 % thresholds, considering





the popularity of the destination selected by the pedestrian agents (among south, east and north) together with their flood risk
state and stability state.

**3.3 Analysis of the results**

Figure 12 shows the total number of evacuating pedestrians in the walkable area, plotted according to the pedestrians' choices
among the three destinations located to the south, east and north of the stadium, for the 20 %, 30 % and 40 % thresholds. All
the simulated evacuation patterns show a decrease in the total number of pedestrians after 22 min of flooding. This suggests
that 22 min would be required for the 4,080 pedestrians to vacate the stadium, and that the choice for the threshold does not
have any effect on the density of pedestrians evacuating.
The simulated evacuation patterns obtained with the 20 % threshold are shown in Fig. 12a, suggesting that most of
the pedestrians evacuated the walkable area within 35 min. The majority of the evacuating pedestrians start favouring the south
destination after 5 min, indicating that after this time the floodwater depth is beyond 20 % of their body height over the
pathways of the walkable area extending towards the east and north destinations. After 5 min, the south destination remained
the most popular destination, selected by more than 55 % of the pedestrians; whereas, the east and north destinations were less
popular, selected by 25 % and 20 % of the pedestrians, respectively.
With the simulated evacuation patterns obtained with the 30 % threshold (Fig. 12b), a longer evacuation time is
predicted for the majority of the evacuating pedestrians. Now it takes about 45 min for most of the pedestrians to leave the
walkable area and the popularity of the east and north destinations increased, with slightly more evacuating pedestrians
preferring them, about 25 % and 25 %, respectively. This suggests that 5 % more of the pedestrians considered changing their
destination to the north where the floodwater depth can only reach up to their knee height. Still, as with the 20 % threshold,
the south destination was the most popular and started to favour after 5 min by 50 % of the pedestrians.
With the simulated evacuation patterns obtained with the 40 % threshold (Fig. 12c), a significant change in the
favoured destination is observed alongside an even more prolonged evacuation time. Now, it takes about 55 min for most of
the pedestrians to evacuate the walkable area and the popularity of the south destination decreased significantly, compared to
the predicted evacuation patterns obtained with the lower thresholds. Here, the south destination was only picked up by 25 %
of the pedestrians and the north destination was preferred instead (by around 50 % of the evacuating pedestrians) since the
very beginning of the evacuation. As for the east destination, it remained equally popular as with evacuation patterns obtained
with the lower thresholds, and was selected by around 20 % of the evacuating pedestrians.
The simulated evacuation patterns in Fig. 12 imply that the south destination would be the preferred by people who
are less likely to enter floodwater with a depth beyond their knee height, and that the north destination would be preferred by
those willing to enter the deeper floodwater. The results also suggest longer evacuation times when people are willing to enter
the floodwater at a depth beyond their knee height.




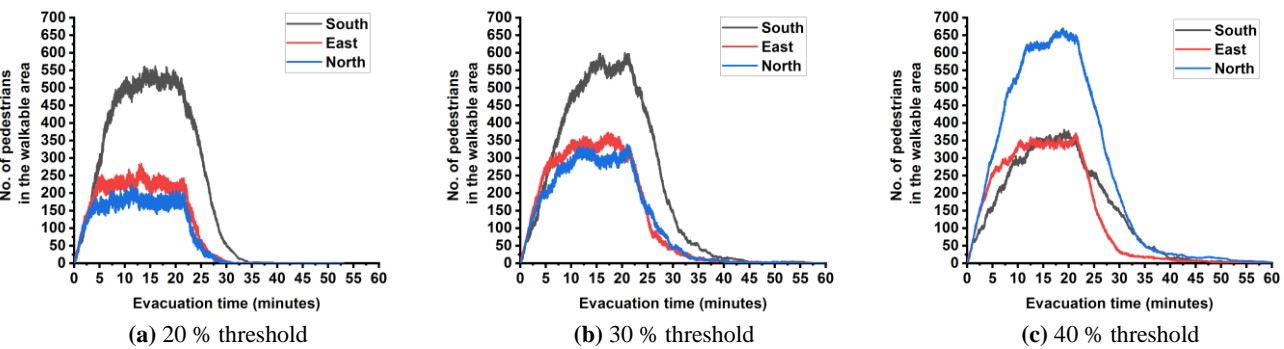

**(a)** 20 % threshold  **(b)** 30 % threshold  **(c)** 40 % threshold

**Figure 12:** Total number of evacuating pedestrians in the walkable area plotted according to their destination choices for the south, east and north during the evacuation time: (a) 20 % threshold, (b) 30 % threshold and (c) 40 % threshold

The flood risk states and stability states with toppling-only or toppling-and-sliding conditions for the pedestrians are shown in Fig. 13 for the three simulations obtained with the 20 %, 30 % and 40 % thresholds. Fig. 13-left includes the flood risk states as well as the total number of evacuating pedestrians in the walkable area. As the threshold increased, the total number of pedestrians in the walkable area is seen to increase, leading to prolonged evacuation times. This observation is aligned with that made for Fig. 12, suggesting that the evacuation process would be delayed as more evacuating pedestrians enter the deeper floodwater where their moving speed reduces. The number of pedestrians in dry zones remains constant, despite the choice for the threshold. This may be expected as these pedestrians represent those who had initially decided to go towards the south destination (one third of the pedestrians) and did not find a need to alter their destination during the process given the dominance of dry areas within south parts of the walkable area (see Fig. 10). For the three thresholds, the majority of the evacuating pedestrians were found to be at a low flood risk state (HR < 0.75). Up to around 20, 100 and 200 evacuating pedestrians reached a medium flood risk state (0.75 < HR < 1.5) with the 20 %, 30 % and 40 % thresholds, respectively. Up to only 15 pedestrians had a high risk flood state (HR > 1.5) and this only occurred for those who entered the floodwater at a depth beyond 40 % of their body height.

The number of evacuating pedestrians that could have a stability state with a toppling-only or toppling-and-sliding conditions are shown in Fig. 13-right. For the 20 % threshold, very few pedestrians were found to have these stability states, up to 15 in number. Findings in Shirvani et al. (2020) suggest that these could be pedestrians with a low flood risk state (HR < 0.75) with a toppling-only condition or with a medium flood risk state (0.75 < HR < 1.5) with a toppling-and-sliding condition. The number of pedestrians with these conditions for the stability states increased with the threshold of 30 %, which is expected given the increased number of pedestrians under a low and medium flood risk state evacuating over a longer period. Up to 60 and 30 more pedestrians were found to be with toppling-only and toppling-and-sliding conditions, respectively. With the 40 % threshold, 20 more pedestrians were found to be with a toppling-and-sliding condition, and up to 200 more were found to be with a toppling-only condition. The significant increase in the number of pedestrians with a topping-only condition





is expected with the 40 % threshold, for which more pedestrians would be entering the floodwater where its depth is beyond
their knee height.

The analysis of the flood risk and stability states suggests that the majority of people evacuating the stadium would

take an evacuation route that is either dry or keeps them under a low flood risk state (HR < 0.75) with a toppling-only condition
during the evacuation. Less people would be entering deeper floodwater and, when they do, they are expected to be in a
medium flood risk state (0.75 < HR < 1.5) where they can have a toppling-and-sliding condition.

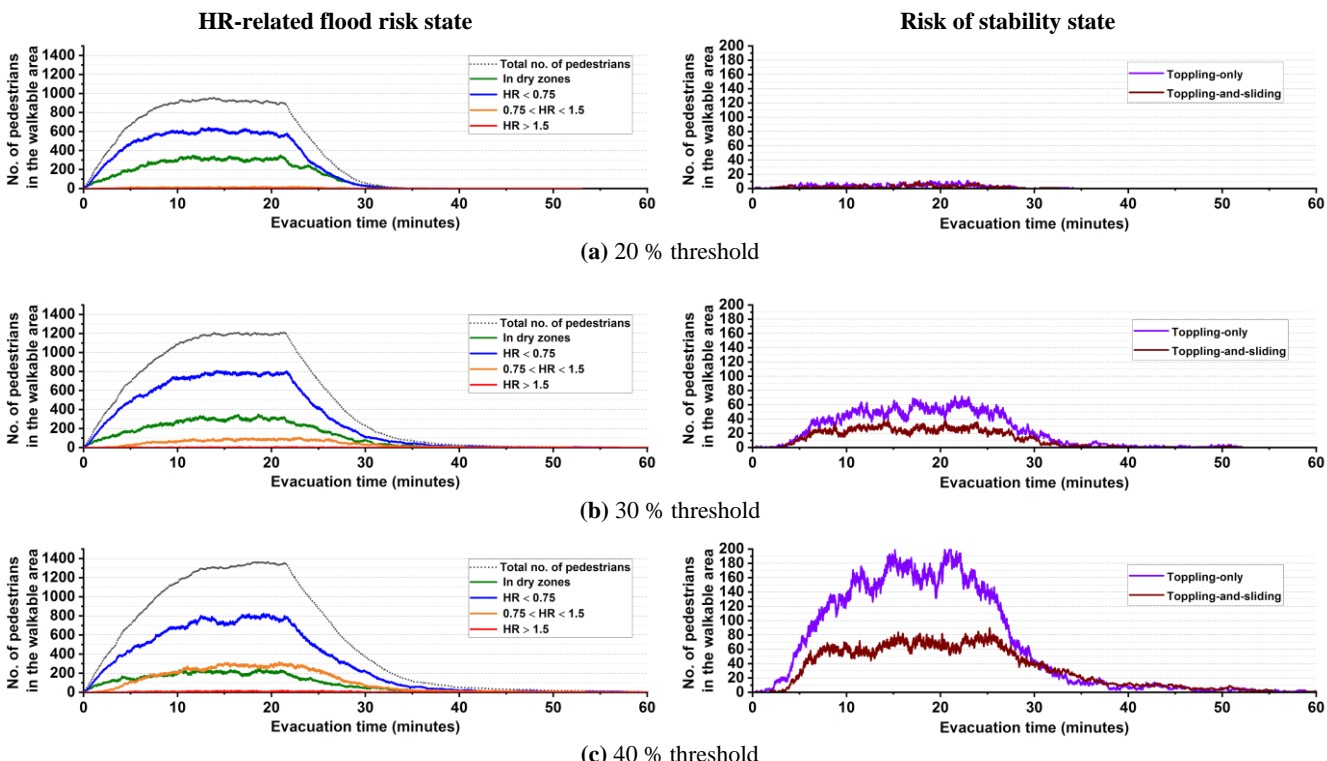

**Figure 13:** Total number of evacuating pedestrians in the walkable area plotted according to their HR-related flood risk state (left panel) and stability state when they were immobilised in floodwater (right panel) during the evacuation time: (a) 20 % threshold, (b) 30 % threshold and (c) 40 % threshold

Figure 14 shows the 2D spatial distribution of the evacuating pedestrians over the HR flood map at 22 min when

pedestrian presence in the walkable area is at its highest as soon as everyone vacated the stadium. The pedestrians are
represented by dots with different colours representing their stability state based on the predictions made with the 20 %, 30 %
and 40 % thresholds. The evacuation patterns in Figure 14, though retrieve many of observations made before (through Fig.
12 and Fig. 13) demonstrate the simulator's further ability to inform on the potential locations where the evacuating pedestrians
are expected to be immobilised by the floodwater. With the 20% threshold (Fig. 14a), most of the pedestrians remained mobile
in the floodwater (stable condition) and preferred the south destination where low flood HR dominates. From the remaining
pedestrians, who preferred the east or north destinations, a handful were immobilised by floodwater (toppling-only or toppling-





and-sliding conditions). These states of immobility are observed to occur at two vicinities where the flood HR varied from the
upper low range to the medium range: within the north-east pathways located at the streets' intersection leading to the north
destination, and within middle of the eastern the pathways leading to the east destination. The spatial distributions predicted
with the 30% threshold (Fig. 14b) also suggest a preference for the south destination by most of the pedestrians, and that many
more pedestrians would be immobilised by the floodwater within the aforementioned vicinities. There, at least a dozen would
have a stability state with a toppling-and-sliding condition caused by the relatively higher number of pedestrians who kept
moving to the north and east destinations. With the 40% threshold (Fig. 14c), most of the pedestrians were still found to be
mobile in floodwater (stable condition) despite the fact that the (riskiest) north destination was the dominant choice. However,
the spatial distributions predicted with this threshold point to a major increase in the number of immobilised pedestrians within
the aforementioned vicinities.
The analysis in Fig. 14 suggests that people who avoid entering a floodwater depth beyond their knee height are most
likely to select the south destination, where their condition remains stable to keep evacuating with minimum risk of
immobilisation. Those with a tendency to enter deeper floodwater would go to the east or north destinations, towards which
the majority would still be able to evacuate, but at a slower pace delayed by the risk of facing immobilisation as they move
forward to their selected destination. Overall, the predictions produced by the simulator (Fig. 12 to Fig. 14) seem sensible,
suggesting that it can be a useful tool for planning people evacuation scenarios in small urban areas to a flood emergency.

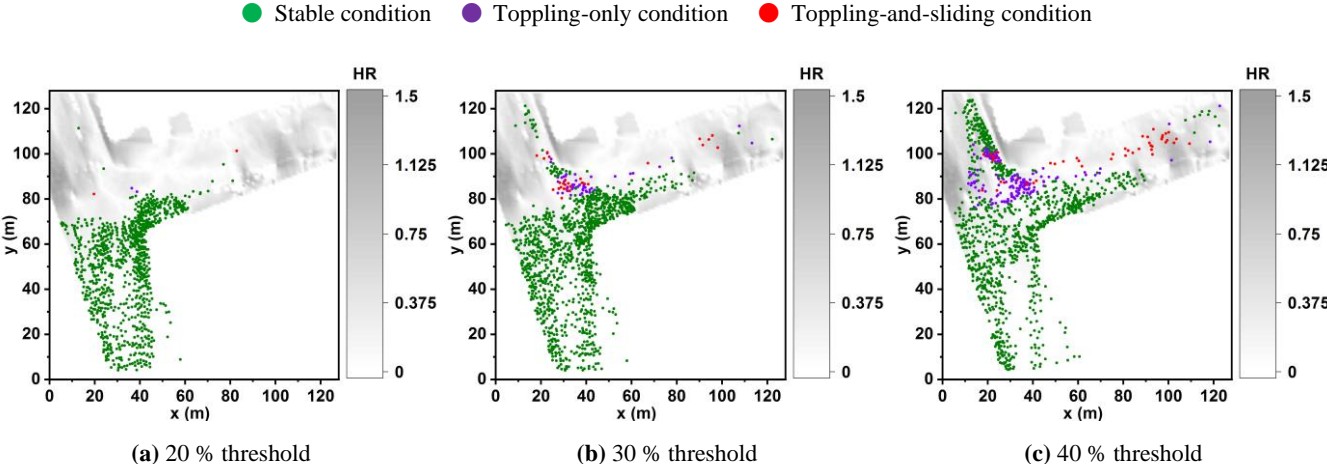

**Figure 14:** The spatial distribution of pedestrians over the walkable area under the predicted stability states (coloured dots) along with the
HR flood map (grey shade) at simulation time $t = 22$ min, when the number of pedestrians over the walkable area is at highest after all of
them had vacated the stadium: (a) 20 % threshold, (b) 30 % threshold and (c) 40 % threshold





## 4 Summary, discussions and limitations

The flood-pedestrian simulator was augmented to more realistically model the dynamics of pedestrian responses around and inside urban floodwaters. The simulator couples a hydrodynamic model to a pedestrian model in a single agent-based modelling platform which allows dynamic exchange and processing of data between the two models as they update in time. The hydrodynamic model is built upon a fixed grid of discrete 'flood' agents, storing and updating information of floodwater in space and time. The pedestrian model also involves a fixed grid of discrete 'navigation' agents that encodes the (urban) features of the walkable area over which continuous type 'pedestrian' agents move in space and time as they interact with each other and make way-finding decisions. The grids of navigation and flood agents are overlapped, allowing the navigation agents to dynamically receive hydrodynamic information to further inform pedestrian agents' way-finding decisions when they move in the floodwater. The pedestrian agents were featured with realistic human body characteristics and in-model behavioural rules based on which their flood risk states, stability states, moving speeds and way-finding decisions are defined at their certain location and output time. The flood risk states were defined using the flood HR quantity, expressed in terms the floodwater depth and velocity magnitude, to be either low, medium, high or highest (Table 1). Experimentally derived formulas were used to define individual stability states based on analysing the floodwater depth and velocity magnitude and specific body height and mass of the pedestrian agents. These states include a stable condition of mobility in the floodwater or immobilisation conditions from toppling-only or toppling-and-sliding (Table 2). The stability states in the present simulator were defined based on more realistic body mass and height that are age and gender related (Table 3). This version of the simulator is also featured by a set of more realistic moving speeds defining the mobility of the pedestrian agents around and inside the floodwater: the social force model was adapted to consider empirically based age- and gender-related walking speeds in dry zones and a maximum excitement condition inducing faster walking speeds around the floodwater (Table 4); and, when inside the floodwater, experimentally derived moving speeds were applied with either a walking condition or a running condition (Table 5). The walking and running conditions can be applied without and with a two-way interaction condition that considers the effects that pedestrian congestion may have on the floodwater dynamics, and vice versa. A new autonomous change of direction condition was added to make the way-finding decisions of pedestrian agents become self-automated (Sect. 2.2.3), informed by the state of the floodwater dynamics or the choices of pathways selected by their neighbouring agents. This condition required introducing a threshold of a floodwater depth to body height beyond which a pedestrian agent decides to change direction and a waiting time after which the pedestrian agent is assigned the most popular pathway selected by the others around.

The conditions factoring in the added characteristics and in-model behavioural rules were evaluated for a test case of a flood evacuation in a shopping centre. This indoor evacuation test case was selected as it was investigated with the previous version of the simulator with which the mobility and stability states of the pedestrian agents were driven by rules without considering age and gender influences, thus allowing to evaluate the added rules and characteristics. The evaluation procedure was based on systematically enabling any of the new age- and gender-related conditions (Table 6), and analysing the changes



they induce in the simulated pedestrian evacuation patterns with reference to the baseline patterns obtained from the previous version of the simulator. The analysis of the simulated pedestrian evacuation patterns indicate major differences with the baseline evacuation patterns using any of the new conditions. Using the walking condition, the simulator predicted much more pedestrians with a low flood risk state exhibiting slower evacuation patterns, and less pedestrians with a medium flood risk state exhibiting faster evacuation patterns before reaching the highest density of the crowd in the walkable area. With the running condition, no major changes in the evacuation patterns were observed compared to those predicted with the walking condition for the pedestrians with a medium flood risk state; however, those with a low flood risk state were predicted to have relatively faster evacuation patterns, even compared to the baseline predictions. With either of the running or walking conditions, only few pedestrians were predicted to be evacuating in a high risk state, which is in contrast with the baseline patterns indicating much more pedestrians to be evacuating. Fewer pedestrians were predicted to be evacuating with the highest risk state, as was also the case with the baseline predictions; however, using either of the running or walking conditions predicted earlier times of highest risk exposure. Choosing either of the running or walking conditions with the two-way interaction condition was noted to accelerate the evacuation patterns for the pedestrians with a low to medium flood risk state. This was particularly observed at flooding time after reaching the highest density of the crowd in the walkable area, leading to more significant reduction in the floodwater hydrodynamics and, in turn, reduction of the state of immobility for many of the pedestrians with a low to medium risk state. Overall, the new features led to different evacuation patterns than the baseline patterns. With the walking condition, the simulator is expected to only lead to major differences in the evacuation patterns of pedestrians in a low flood risk state. This condition is a sensible choice when applying the simulator to study a mass evacuation together with the two-way interaction condition, which could significantly affect the evacuation patterns when there is crowding of pedestrians in low to medium flood risk state.

The utility of the simulator, with the new autonomous change of direction condition, was then demonstrated over a real-world case study of a mass evacuation of football spectators attending a match into a T-junction streetscape, in response to a flood emergency. The study site included the streetscape and the entrance of Hillsborough football stadium in Sheffield, and is prone to flooding that would expose pedestrians to low-to-medium flood risk states. The streetscape leads to three ends, towards the north, east and south, any of which could be a designation for pedestrians. The simulator was set up to replicate historical extents and depths of the floodwater that would inundate this study site, to consider the age- and gender-characteristics of football spectators, and by enabling the walking condition together with the two-way interaction condition. The autonomous change of direction condition was applied based on three thresholds of a floodwater depth to body height: 20% threshold, 30% threshold and 40% threshold, representative of a high, medium and low level of people's risk perception. The simulated evacuation patterns suggest that when people exhibit high to medium risk perception, avoid zones of floodwater depth beyond their knee height, the majority change direction to go to the south destination that has the highest portion of dry zones. Whereas, when people exhibit a low risk perception, enter zones of floodwater depth slightly higher than their knee height, the majority would take the shallowest pathway leading to the north destination. As the risk perception level decreased, the simulated evacuation patterns showed an increase in the number of people in a medium risk state with an immobilised



748 condition and longer evacuation time. The investigations over the real-world case study demonstrates that the flood-pedestrian

749 simulator can be usefully used to analyse the dynamics of people's responses in and around the floodwater as part of the flood

750 risk analysis, thus is a useful tool for planning evacuation of crowds to flood emergencies in small and potentially congested

751 urban areas.

752  However, the flood-pedestrian simulator has a number of considerations and limitations that are worth mentioning.

753 Firstly, the simulator requires the accessibility to a Graphical Processing Unit (GPU) card and the generation of input files

754 requires special .xml translation specific to FLAMEGPU and using the FGPUGridNavPlanEditor toolkit (available at:

755 https://github.com/RSE-Sheffield/FGPUGridNavPlanEditor). Secondly, the simulator can provide a live visualisation showing

756 hydrodynamic and pedestrian information changing in real time, when run on windows using the console mode (Shirvani,

757 2021). Thirdly, in terms of pedestrian characteristics, the simulator does not incorporate the uncertainties associated with social

758 and psychological characteristics of people, e.g. flood tourism, as well as their floating and sinking conditions. Lastly, but not

759 least, the assumptions and thresholds used to implement the two-way interaction condition and the autonomous change of

760 direction condition are both lacking any existing empirical evidence base supported by dedicated laboratory experiments.

761 **Code availability**

762 The flood-pedestrian simulator is accessible from Zenodo open-access repository at https://doi.org/10.5281/zenodo.4564288,

763 with a link to the GitHub source codes of the latest release, including a detailed 'run guide' and input files to enable the users

764 to run the flooded shopping centre and the Hillsborough stadium evacuation test cases on their own machine. The previous

765 version of the simulator is also available on DAFNI, available at: https://dafni.ac.uk/project/flood-people-simulator/, where it

766 can be run from a user-friendly graphical interface and supported by a run guide.

767 **Data availability**

768 Outputs of the simulations are available in the Zenodo open-access repository at https://doi.org/10.5281/zenodo.4576906.

769 **Video supplement**

770 Demo videos of the test cases are available online in the TIB AV-Portal at https://doi.org/10.5446/51547.

771 **Author contribution**

772 MS contributed to developing the simulator, design of the test cases, running simulations, obtaining outputs and figure

773 preparation. GK proposed the research approach and supervised the development, testing, scenario configurations and analysis



of the outputs, and obtained the research grant. MS and GK prepared the manuscript. All authors read and approved the final
paper.

**Competing interests**

The authors declare that they have no conflict of interest.

**Acknowledgements**

This work is part of the SEAMLESS-WAVE project (SoftwarE infrAstructure forMulti-purpose fLood modElling at variouS
scaleS based on WAVElets), which is funded by the UK Engineering and Physical Sciences Research Council (EPSRC) grant
EP/R007349/1. For information about the SEAMLESS-WAVE project visit https://www.seamlesswave.com. The authors wish
to thank Paul Richmond and the Research Software Engineering (https://rse.shef.ac.uk/) group for providing technical support
during the implementation of the flood-pedestrian simulator on FLAMEGPU.

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
