# Peer review of "Flood-pedestrian simulator for modelling human response dynamics during flood-induced evacuation: Hillsborough stadium case study"

_Natural Hazards and Earth System Sciences, 2021_

## Author Comment (AC1)

Dear Editor and Referee #1,

We thank you for the time spent to evaluate and review our manuscript and for the critical and constructive comments. The major and minor comments brought by Referee #1 were all addressed in a revised manuscript.

Here, we will repeat the reviewer's comments (italic font) and response directly below (standard font). After each response, the associated changes applied to the revised manuscript are included in a table box (coloured in red).

Sincerely, G. Kesserwani and M. Shirvani

**General comments**

"The value of Agent-Based Modelling as a technique to understand the role of individual decision making in response to hazards, such as flooding, has increased in recent years, since the initial publication of ABM evacuation behaviour was published in 2011. This study is an interesting example of further development of this application of ABM and demonstrates a good synthesis of empirical research to formulate agent behaviour. Through introducing agent characteristics such as age, gender, weight, variable movement speeds and stability patterns, the model introduces an enhanced level of heterogeneity amongst the agent population from previous iterations of the model."

**Major comments**

"However, there lies two fundamental concerns related to the outputs of the model:"

The two fundamental concerns have been insightful for us to carry out the revision work necessary to ascertain the credibility of the model outputs based on which the results were analysed. The minor/technical comments raised were also found valuable to sharpen the quality of the manuscript. A point-by-point reply to each comment is provided below it, together with an explanation of the associated changes made to the manuscript.

"1. It would appear that uncertainty inherent within modelling stochastic agent behaviour is not acknowledged. The impression given is that model simulations were ran only once for each scenario (lines 331 - 333) and outputs were then analysed. There should be evidence of multiple simulations for each test scenario to account for variance in behaviour of agents. For example, confidence intervals and averaging over an appropriate number of simulations would provide a more representative set of outputs that account for stochasticity."

The reviewer is correct about this important point. As the outputs analysed in the initially submitted preprint were extracted from a single run, they do not account for the uncertainty inherent within stochastic agent behaviour. Therefore, the preprint has been revised to: first, acknowledge this uncertainty; then, to explore the sample size needed for a series of probabilistic runs to reach statistically significant outcomes for a range of confidence levels; and, finally, reproduce new outcomes averaged out from the probabilistic runs for the two case studies explored and each of their test scenario.

Section 2.3.1 was therefore extended to:

...As the motion of each pedestrian agent is governed by a stochastic (space-time) process, series of 10 and 20 simulation runs were conducted to average out a plausible outcome for each of the configuration Modes. The plausibility of the average outputs from both series of runs is evaluated, by estimating the margin of error (*MOE*) assuming confidence levels ranging between 90% and 99.9%. The following formula is used to evaluate the *MOE*:

$$MOE = Z_{score} \times \sqrt{\frac{\sigma^2}{n}}, \tag{3}$$

where,  $Z_{score}$  is the critical value, which is equal to 1.65, 1.96, 2.17, 2.58 and 3.29, for confidence levels of 90 %, 95 %, 97 %, 99 % and 99.9 %, respectively (Hazra, 2017);  $\sigma$  is the standard deviation from the sample of outputs of size

 $n = \{10, 20\}$ ; and  $\sigma = \sqrt{\frac{\Sigma(x_i - \overline{x})^2}{n}}$ , with  $x_i$  representing the number of pedestrians with a particular HR-related flood risk or stability state extracted from the recorded outputs, and  $\overline{x}$  is the averaged value. Table 7 lists the maximum *MOE* evaluated for the different confidence levels, with respect to the average number of pedestrian agents under different HR-related flood risk and stability states for configuration Mode 0 to Mode 4.

**Table 7:** Maximum margin of error (*MOE*) for the average number of pedestrian agents with different HR-related flood risk or stability states that are extracted from the recorded outputs of all the configuration modes (Table 6) and across different confidence levels ranging from 90 % to 99.9%. Different ranges of the evaluated maximum *MOE* are highlighted with different colour shades: green, orange and red to indicate  $MOE \le \pm 5$ ,  $6 \le MOE \le 9$  and  $MOE \ge 10$ , respectively.

|      | HR-related flood risk and | Maximum MOE |      |      |             |        |      |      |      |             |        |
|------|---------------------------|--------------------|------|------|-------------|--------|------|------|------|-------------|--------|
| Mode |                           | n = 10      |      |      |             | n=20   |      |      |      |             |        |
|      | stability states          | 90 %               | 95 % | 97 % | 99 % | 99.9 % | 90 % | 95 % | 97 % | 99 % | 99.9 % |
|      | HR < 0.75                 | ± 5                | ± 6  | ± 6  | ± 8         | ± 10   | ± 3  | ± 4  | ± 4  | ± 5         | ± 6    |
| 0    | 0.75 < HR <1.5            | ± 4                | ± 5  | ± 6  | ± 7         | ± 9    | ± 3  | ± 3  | ± 4  | ± 4         | ± 5    |
|      | 1.5 < HR <2.5             | ± 3                | ± 3  | ± 4  | ± 5         | ± 6    | ±2   | ± 3  | ± 3  | ± 4         | ± 5    |
|      | HR > 2.5                  | ± 1                | ± 1  | ± 1  | ± 2         | ±2     | ± 1  | ± 1  | ± 1  | ± 1         | ± 1    |
|      | Toppling-only             | ± 5                | ± 6  | ± 7  | ± 8         | ± 10   | ± 4  | ± 4  | ± 5  | ± 6         | ± 8    |
|      | Toppling-and-sliding      | ± 4                | ± 5  | ± 5  | ± 6         | ± 8    | ± 3  | ± 4  | ± 4  | ± 5         | ± 9    |
|      | HR < 0.75                 | ± 6                | ± 7  | ± 8  | ± 9         | ± 12   | ± 4  | ± 4  | ± 5  | ± 6         | ± 7    |
|      | 0.75 < HR <1.5            | ± 6                | ± 7  | ± 8  | ± 10        | ± 12   | ± 4  | ± 4  | ± 5  | ± 6         | ± 7    |
|      | 1.5 < HR <2.5             | ± 1                | ± 1  | ± 1  | ± 1         | ± 1    | ± 1  | ± 1  | ± 1  | ± 1         | ± 1    |
| 1    | HR > 2.5                  | $\pm 0$            | ± 0  | ± 0  | ± 0         | ± 1    | ± 0  | ± 0  | ± 0  | ± 0         | ± 0    |
|      | Toppling-only             | ± 6                | ± 8  | ± 8  | ± 10        | ± 13   | ± 4  | ± 4  | ± 5  | ± 6         | ± 7    |
|      | Toppling-and-sliding      | ± 5                | ± 6  | ± 7  | ± 8         | ± 10   | ± 3  | ± 4  | ± 5  | ± 5         | ± 7    |
|      | HR < 0.75                 | ± 6                | ± 7  | ± 7  | ± 9         | ± 11   | ± 4  | ± 5  | ± 5  | ± 6         | ± 8    |
|      | 0.75 < HR <1.5            | ± 7                | ± 8  | ± 9  | ± 10        | ± 13   | ± 5  | ± 6  | ± 6  | ± 7         | ± 9    |
| 2    | 1.5 < HR <2.5             | ± 1                | ± 1  | ± 1  | ± 2         | ±2     | ± 1  | ± 1  | ± 1  | ± 1         | ± 1    |
|      | HR > 2.5                  | ± 1                | ± 1  | ± 1  | ± 1         | ± 1    | ± 1  | ± 1  | ± 1  | ± 1         | ± 1    |
|      | Toppling-only             | ± 6                | ± 7  | ± 8  | ± 9         | ± 12   | ± 4  | ± 4  | ± 5  | ± 6         | ± 7    |
|      | Toppling-and-sliding      | ± 6                | ± 7  | ± 7  | ± 9         | ± 11   | ± 4  | ± 5  | ± 5  | ± 6         | ± 8    |
|      | HR < 0.75                 | ± 6                | ± 7  | ± 8  | ± 9         | ± 12   | ± 4  | ± 4  | ± 5  | ± 6         | ± 7    |
|      | 0.75 < HR <1.5            | ± 6                | ± 7  | ± 8  | ± 9         | ± 12   | ± 4  | ± 5  | ± 5  | ± 6         | ± 8    |
| 3    | 1.5 < HR <2.5             | ± 1                | ± 1  | ± 1  | ± 1         | ± 1    | ± 0  | ± 1  | ± 1  | ± 1         | ± 1    |
|      | HR > 2.5                  | $\pm 0$            | ± 0  | ± 0  | ± 0         | ± 0    | ± 0  | ± 0  | ± 0  | ± 0         | ± 0    |
|      | Toppling-only             | ± 6                | ± 7  | ± 8  | ± 9         | ± 12   | ± 4  | ± 5  | ± 5  | ± 6         | ± 8    |
|      | Toppling-and-sliding      | ± 6                | ± 7  | ± 8  | ± 9         | ± 12   | ± 4  | ± 4  | ± 5  | ± 6         | ± 7    |
| 4    | HR < 0.75                 | ± 5                | ± 6  | ± 7  | ± 9         | ± 11   | ± 4  | ± 4  | ± 5  | ± 6         | ± 7    |
|      | 0.75 < HR <1.5            | ± 7                | ± 9  | ± 10 | ± 12        | ±15    | ± 5  | ± 6  | ± 6  | ± 7         | ± 10   |
|      | 1.5 < HR <2.5             | ± 1                | ± 1  | ± 1  | ± 1         | ±2     | ± 1  | ± 1  | ± 1  | ±1          | ± 1    |
|      | HR > 2.5                  | ± 1                | ± 1  | ± 1  | ± 1         | ± 1    | ± 0  | ± 0  | ± 1  | ± 1         | ± 1    |
|      | Toppling-only             | ± 6                | ± 7  | ± 8  | ± 9         | ± 12   | ± 4  | ± 4  | ± 5  | ± 6         | ± 7    |
|      | Toppling-and-sliding      | ± 6                | ± 7  | ± 8  | ± 9         | ± 12   | ± 4  | ± 5  | ± 5  | ± 6         | ± 7    |

For n = 10, there is a considerable increase in the maximum *MOE* with Mode 1 to Mode 4 compared to Mode 0. This is particularly seen for the number of pedestrian agents in low and medium flood risk states (HR < 0.75 and 0.75 < HR <1.5, respectively) and with toppling-only and toppling-and-sliding stability states. This suggests that the more sophisticated the pedestrian agent characteristics and rules, more discrepancies would appear in the simulator's outcomes. The maximum *MOE* identified suggests a deviation of around  $\pm$  15 from the averaged outcomes. However, when the sample size is increased to n = 20, the maximum margin of error does not exceed  $\pm$  10 for all the modes and confidence levels. Therefore, the simulation results analysed next are averaged out from a sample of 20 simulation runs, subject to  $\pm$  10 maximum *MOE* for a population of 1000 pedestrians in the flooded walkable area, which corresponds to 1 % per 1000 pedestrian population.

With the changes above, Sect. 2.3.2 has been revised too in order to incorporate new figures and accommodate their respective changes in the text.

Section 3.2.3 has also been revised to confirm the appropriateness of using a series of 20 simulation runs to average out outputs for the real-case study:

**3.2.3 Simulation runs**

Series of 20 simulations were run under configuration Mode 2, which was deemed suited for the scope of this case study (Sect. 2.3.2). Three series were performed associated with the 20 %, 30 % and 40 % thresholds for the 'autonomous change of direction' condition, respectively (visualisation of a simulation can be found in the video supplement in Shirvani (2021)). Outputs averaged from each series of simulations included spatial and temporal information, at each time step, about the pedestrian agents as they evacuate (t > 0 min). The averaged outputs include the position, HR-related flood risk state, stability state (with a toppling-only condition and a toppling-and-sliding condition), and the choice for the destination selected by the pedestrian agents during the evacuation process. Considering the uncertainties associated with the motion of pedestrian agents, the plausibility of the averaged outputs relevant to the number of pedestrian agents was evaluated. The evaluation was based on the *MOE*, using Eq. (3), for 99.9 % confidence level only, informed by the results of the analysis in Sec. 2.3.1. Table 8 shows the maximum *MOE*s found for the number of pedestrians predicted to be in the considered HR-related flood risk states and the stability states, obtained from the series of runs using each of the 20 %, 30 % and 40 % threshold, respectively. It can be seen that the maximum *MOE* increases as the risk perception level decreases, suggesting a notable increase in uncertainty from the incorporation of the perception component on the pedestrian behaviours.

**Table 8:** Maximum margin of error (*MOE*) for the average number of pedestrian agents with different HR-related flood risk or stability states that are extracted from the recorded outputs throughout the simulations for each 20 %, 30 % and 40 % threshold.

| HR-related flood risk | Maximum MOE |                |                |  |  |  |  |
|-----------------------|--------------------|----------------|----------------|--|--|--|--|
| and stability states  | 20 % threshold     | 30 % threshold | 40 % threshold |  |  |  |  |
| HR < 0.75             | ± 16               | ± 16           | ± 19           |  |  |  |  |
| 0.75 < HR <1.5        | ± 2                | ± 8            | ± 15           |  |  |  |  |
| HR > 1.5              | $\pm 0$            | ± 1            | ± 2            |  |  |  |  |
| Toppling-only         | ± 2                | ± 5            | ± 13           |  |  |  |  |
| Toppling-and-sliding  | ± 1                | ± 4            | ± 7            |  |  |  |  |

Next, the averaged outputs are analysed for each of the 20 %, 30 % and 40 % thresholds, considering the popularity of the destination selected by the pedestrian agents (among south, east and north) together with their flood risk state and stability state.

With the changes above, Sect. 3.3 have been revised too in order to incorporate new figures and accommodate their respective changes in the text.

"2. The number of agents included in the Hillsborough case study is 4,080. The lowest recorded attendance for Sheffield Wednesday in 2019 was counted at 21,485, meaning that the agent population is only 18% of what is considered 'real'. Given that the start of the paper states that the FLAMEGPU platform can handle "as large population size as needed (line 79)" and that throughout the paper congestion is frequently stated as a factor that influences flow dynamics and evacuation time, a question emerges as to why such a significantly low agent population has been adopted. The paper states that "using a bigger population size would lead to extreme pedestrian congestion that impacts the movements of individual pedestrians (lines 489 – 490)". Surely, in the context of a football match, this is an important factor that must be represented as accurately as possible. There is no justification for the chosen figure of 4,080 and it seems arbitrary. I am not wholly convinced that subsequent outputs reflect evacuation times that would be representative of a football match. Either a more realistic agent population in line with actual attendance rates is necessary, or justification that the emergent behaviour is not dramatically impacted by the chosen number of spectators."

The reviewer raised fair points regarding the lack of justification of the selected 4,080 agent population and of how effects of pedestrian congestion on the evacuation times were handled. In terms of agent population, the average recorded attendance should be around 24,000 in normal weather conditions. As the case study assumes severe weather conditions and flood warnings were issued before the event, it makes sense to consider around 20% of the expected spectators; this is also representative of the percentage of people who would ignore the warnings and attend the match (Fielding et al., 2007).

As the statement: "using a bigger population size would lead to extreme pedestrian congestion that impacts the movements of individual pedestrians" was a major oversight, it has been removed from the manuscript. Instead, new clarifications are provided on how a dischach rate was implemented to the simulator for this case study to comply with a safe evacuation measure, in other words ensure that "the emergent behaviour is not dramatically impacted by the chosen number of spectators".

End of Sect. 3.1 has been extended to more clearly justify the reduced number of spectator selected and to also justify the outcomes of the simulator won't be significantly impacted for a higher population size.

In this study, it is assumed that the site in Fig. 7 is hit by a flood, similar to the one that had happened in 2019, during a football match where the spectators are caught unaware of the rainfall accumulation around the stadium. ... The evacuated spectators gradually enter the walkable area where they get in direct contact with the propagating floodwater along their ways to any of the south, east or north destinations.

A population of 4,080 spectators was assumed, which is lower than normal due to the severe weather condition and flood warnings issued prior to the event. This population is around 20 % of the spectators expected, and represents the relative number of people who would ignore the warnings and attend the match (Fielding et al., 2007). For this case study, a dispatch measure was introduced to the simulator to release the evacuees into the walkable area during the flooding. The dispatch measure limits the influx rate to person-per-second per width unit to comply with guidance methods for controlling the density of large crowds outside the stadiums for safe evacuation (Minegishi and Takeichi, 2018; Still, 2019). For a gate that is around 4 m wide, four pedestrians per second are dispatched from the stadium to the walkable area. Using the simulator with this dispatch rate limits the overall number of pedestrians that would be present in the walkable area at a time. Therefore, running the simulator to analyse the evacuation of a larger number of spectators is expected to lead to similar evacuation patterns and flood risk trends that would be prolonged over a larger evacuation time.

The flood-pedestrian simulator is applied to simulate this scenario for analysing the pedestrian evacuation patterns including their preference for the destination during flood evacuation, by activating the 'autonomous change of direction' condition (Sect. 2.2.3).

**Technical/Minor Comments:**

"In section 2.3.2, it states "...the more crowding of pedestrians the more energy loss in the floodwater dynamics for low risk floodwaters, which in turn enables the pedestrians located behind to take a faster moving speed (lines 346 – 348)". Whilst it is well noted that crowding would disperse the floodwater, making pedestrian movement easier, I can't help but wonder if collective moving speed would in turn be slowed down by the congestion itself, therefore this could be acknowledged."

While revising Sect. 2.3.2, the above-raised aspect has been addressed to acknowledge the influence of pedestrian congestions in the floodwater on the collective moving speed. Our findings suggest that, with the two-way interaction condition, crowding of pedestrians disperse the floodwater, in turn allowing more spreading of the moving pedestrians and those in the front to move faster (see revised Fig. 6 below). However, the collective moving speed does not exhibit any significant difference due to congestion, as can also be noted in the overall trends of HR-related flood risk states of pedestrians (addressed in the discussion of the revised Fig. 4, not shown in this letter).

The discussion of Fig. 4a in Sect. 2.3.2 was revised to acknowledge this aspect:

Figure 4 shows the trends in the number of evacuating pedestrians with different HR-related flood risk states predicted by the simulator after 20 runs using all the configuration modes (Table 6). Figure 4a represents how the number of pedestrians with a low flood risk state (HR

"The model demonstrates well a fundamental concept inherent within complexity sciences and Agent-Based Modelling; emergence. The adoption of risk perception thresholds seem to have influenced the favouring of destination selection by agents with varying levels of risk perception. As is the case for risk perception and adaptive behaviour in response to flooding (i.e. risk perceptions thresholds increase after an individual's property is flooded, which increases the likelihood of adopting protective measures in future) as an active area of interest within flood risk management. Some further comments on the overlap between risk perception and evacuation behaviour may be insightful to lay foundations for future research."

In the revised paper, comments have been added to elaborate on the overlap between risk perception and evacuation behaviour, namely supported by further citing and discussing recently published papers who studied and reviewed the impact of past flooding experience of individuals on behaviours. Also, the uncertainty analysis added to Sect. 3.2.3 (to address major comment #1 above) offers new insights on the impact of adding the risk perception component on the variance in the simulator outcomes.

**2.2.3 Autonomous change of direction condition**

Each pedestrian agent is also featured with two extra rules to enable it to autonomously navigate into new pathways while moving within a flooded zone, where it encounters a non-zero floodwater depth from the navigation agent at its specific time and location. The first rule makes a pedestrian agent detect and choose another destination if the floodwater depth along its way becomes higher than a threshold of a floodwater depth to body height. The choice for the threshold is case-dependent and exploring different thresholds may be necessary (Sect. 3.3.2) as an individual's flood risk perception is dependent on different factors, including past flooding experiences (Hamilton et., 2020; Abebe et al., 2020). This affects the modelling of decisions of when and where people enter the floodwater or make a move into another destination (Becker et al., 2015; Netzel et al., 2021).

Also, the following changes are made in the 3rd paragraph of Sect. 3.2.2.

As people's behaviour in floodwater is uncertain depending on their individual risk perceptions (Sec. 2.3.2), they may take the risks and enter the floodwater. To accommodate this uncertainty in the simulator, three thresholds of floodwater depth to body height were investigated (Fig. 11), inspired by the experiments in Dias et al. (2021). The '20 % threshold' was defined to represent people with high-risk perception, who previously experienced a critical flooding incident, and decided not to enter floodwater with a depth that is more than 20 % of their body height. This threshold, for this case study, is estimated based on the ratio of the dominant minimum value for the depth of floodwater that can occur over the walkable area (0.3 m) to the height of the shortest pedestrian agent available (1.4 m). With this threshold, the likelihood of the entire population to be in a condition to change their direction is ensured. The '40 % threshold' was defined to represent people with low-risk perception, who previously experienced a minor flooding incident, and decided to enter a floodwater with a depth that is even more than 40 % of their body height. This threshold, for this case study, is estimated based on the ratio of the dominant maximum depth of floodwater (0.9 m) to the height of the tallest population of pedestrian agents available (2.1 m). This threshold enables the entire population to have the freedom to keep moving even within the deepest floodwater in the walkable area (0.9 m). The '30 % threshold' was defined based on an average-risk perception, for those who previously experienced a major flooding incident. Pedestrians with average-risk perception would decide to enter floodwater up to their knees, constituting 30 % of human body height (Teichtahl et al., 2012).

Recall also the above-reported revisions applied to Sect. 3.2.3, addressing major comment #1, which demonstrates the impact of the risk perception on the variability in the simulator outcomes.

• "Some more clarity on what solvers are used to dictate the flow of water in the hydrodynamic model should be provided in Section 3.2.1."

Clarification has been provided in the second paragraph of Sect. 2.1 instead, as it is the most appropriate location to include this general information about the simulator's make-up.

Changes made to Sect. 2.1:

The environment layout encodes force vector fields providing navigation to key destinations. These fields are stored within a grid of fixed discrete agents, forming a navigation map (Karmakharm et al., 2010). The navigation map is necessary for pedestrians' way-finding decisions while they are directed to reach their key destinations.

The hydrodynamic model is formulated based on a non-sequential implementation of a finite volume solver of the depth-averaged shallow water equations on a two-dimensional grid on FLAMEGPU, which was validated previously in Shirvani et al. (2021). The hydrodynamic model was applied on another fixed grid of discrete agents, flood agents, which is coincident with the grid of navigation agents...

Calibration/validation evidence of the hydrodynamic model should be provided in more detail to highlight the robustness of hydrodynamic outputs. Two separate figures are provided of flood extents; an Environment Agency figure (Figure 8) and your own (Figure 10). These could be combined into one figure, by overlaying and analysing to provide an FSTAT value to indicate accurate representation of simulated outputs versus simulated. Similarly (at a minimum), a figure showing simulated and observed hydrographs should be provided to demonstrate that flood depths are within an acceptable tolerance."

To the best of our knowledge, there is no record of an observed hydrograph at a gauge point located in the selected study site, and that's why we resorted to the flood risk maps of the Environment Agency (EA). These flood risk maps inform the range of possibilities of what the flood depth and extent would be for different return periods (Fig. 8), whereas the hydrodynamic model predicts maps and depth extents (Fig. 10) caused by the rainfall runoff from the events of 19th of Nov. 2019 (driven by the hydrograph in Fig. 9). Therefore, we had to keep them separated. Still, the range of water depth and extent predicted by the hydrodynamic model for this event agree well with the expected ranges sampled in the EA flood risk maps should a flooding occur under any of the reported scenarios, which is a good sign that the hydrodynamic model is fairly calibrated for the intended purpose: demonstrating the simulator's capability for modelling collective trends of flood risk states of pedestrians inferred from the heterogeneity in their characteristics and behaviours.

The following footnote text has been added as a footnote to Sect. 3.2.1 to provide clarification:

To the best of the authors' knowledge, there is no record of an observed hydrograph sampled at a gauge point located in the selected study site, though resorting to the EA's flood risk maps can be deemed sufficient to support the scope of this investigation that is mainly focused demonstrating the simulator's capability for modelling collective evacuation trends inferred from the heterogeneity in pedestrian characteristics and behaviours.

The '*Flood-pedestrian simulator user guide*' file on Zenodo (10.5281/zenodo.4672125) includes a summary of all agent variables and values upon initialisation, and many other informative steps to promote replicability and reproducibility.

• "A supplementary table providing a summary of all agent variables and values upon initialisation (flooding, navigation and pedestrian agents) would be useful. This would promote replicability and reproducibility."

---

## Author Comment (AC2)

Dear Editor and Referee #2,

We thank you for the time spent to evaluate and review our manuscript and for the review comments and discussions. Answers to the issues raised by Referee #2 are provided below, and have been addressed in a revised manuscript.

Here, we will repeat the reviewer's comments (italic font) and response directly below (standard font). After each response, the associated changes applied to the revised manuscript are included in a table box (coloured in red).

Sincerely,
G. Kesserwani and M. Shirvani
* * *
*" The paper "Flood-pedestrian simulator for modelling human response dynamics during flood-induced evacuation: Hillsborough stadium case study" aims at showing the effect of human-body characteristics and in-model behavioural rules when included in an ABM integrated framework for flood evacuation modelling. The model is largely based on the previous version developed by the authors, and it is tested on a synthetic case study and on a real-world case. The topic of this study is definitely highly relevant and timely. However, I think that the study could be further strengthened by assessing the effect of the evacuation time and characteristics of the inflow hydrograph on the evacuation process. These factors could have a higher impact than the human-body characteristics considered by the authors for flood risk mitigation purposes."*

Thank you for highlighting the timeliness and relevance of the topic investigated. Before we respond to the comments, we clarify that the new version of the simulator is augmented by empirically-derived behavioural rules to particularly: account for the realistic heterogeneity in pedestrian agents' characteristics (human age, gender and body mass) and behaviours (mobility states, risk perception thresholds); and, their back interaction on the floodwater dynamics. In the context of microscopic evacuation modelling in small urban areas, both the heterogeneity and the back interaction factors could play a determinant role to assess during-flood risk to people and the potential changes in their spatial and temporal evacuation patterns. In this context, the scope of this paper has been to explore the impact of pedestrian characteristics and behaviours on evacuation timing and risk assessment, and not the evacuating timing on general risk assessment.

In the revised introduction (3$^{rd}$ paragraph, Sect. 1), the scope of study has been clarified, while also acknowledging the importance of evacuation time and characteristics of the inflow hydrograph as the reviewer suggested:

… These ABM-based evacuation simulation tools were developed to inform emergency plans for severe flood types, such as in the immediate aftermath of a dam-break or a tsunami wave (e.g. Lumbroso et al., 2021). The focus of these tools is mainly on estimating the loss of life, pinpointing bottlenecks and high-risk areas, and assessing how flood warnings of an impending flash flood could reduce the number of casualties and injuries. For this type of risk analysis, individuals' microscopic decisions and actions are considered insignificant in influencing the overall simulation outcomes due to the scale and speed of floodwater flow. However, for the most common flood types in urban areas, e.g. surface water due to extreme rainfall, less attention has been given to model the microscopic responses, down to the scale of the moving individuals, in and around flooded urban hubs (Ramsbottom et al. 2006). In this context, Bernardini et al. 2021 imported outputs of a flood model into a commercially available evacuation modelling tool, called MassMotion, to analyse flood risk differences in microscale and macroscale modeling with and without including pedestrians' microscopic evacuation behaviour. They concluded that incorporating pedestrians' microscopic evacuation behaviour in microscale modelling could significantly influence the spatial and temporal changes in flood risk to people, i.e. up to 15 % in absolute terms, when compared to macroscale modelling. Their findings also suggest the need to further incorporate non-homogeneous characteristics of people in a more flexible microscale modelling framework, which may result in additional differences to the analysis of flood risk to people.

Having clarified that, a point-by-point reply to each comment raised is provided below it, together with an explanation of the associated changes made to the revised manuscript.

- *" It has been recently shown that the timing in which the evacuation is issued is crucial for reducing flood risk (Alonso et al., 2020). However, in section 2.3.1 the authors state that "When the floodwater starts to propagate over the walkable area, simulation time (t) of 0 min, the pedestrian agents start the evacuation ..." In a no-flooding situation, agents are randomly moving based on their behaviours and on their daily routines. Why an agent should start moving exactly when the water starts entering the building and not before or after? Are there supporting evidence to justify such an assumption? Then, when flooding occurs, there can be two extreme situations. On the one hand, if the agent is doing something else it may not notice the flooding until the pedestrian agent goes on a flooded area. On the other hand, the agent could be informed earlier about a coming flood and start to evacuate earlier. These two scenarios could have dramatic consequences. I suggest including more modes (table 6) accounting for different evaluation timing. This study could show the role of human-body characteristics and in-model behavioural rules in reducing flood risk when evacuation is issued late. "*

To evaluate the relevance of newly-added pedestrian agent's human-body characteristics and in-model behavioural rules, we used Take a Previous model and Add Something (TAPAS) approach, as justified in the initially submitted preprint (Sect. 2.3). To do that, we reconsidered the same synthetic test case of the shopping centre (Sect. 2.3.1) under the same flooding condition as investigated before in the previous version of the simulator (Shirvani et al., 2020). This test case is designed to evaluate the simulator in capturing (spatially and temporally) the two-way dynamic interactions between the pedestrians and the floodwater flow. Therefore, the evacuation was set to happen at the onset of flooding to produce the most critical scenario that would cause the maximum interactions between moving pedestrians and flowing floodwater. This assumption is based on the fact that pedestrians are mainly reliant on their own visual detection of the upcoming hazard and/or an immediate announcement in such a small urban hub. In addition, the selected evacuation timing allows us to analyse the relevance of the added characteristics and rules across a full range of flood risk HR states. This means that further exploring pre-flood or post-flood evacuation scenarios, while feasible, won't allow us to assess the potential impacts neither from the added rules and characteristics nor from the during-flood interactions.

- *" My second concern relates to the shape of the hydrograph considered in the synthetic experiment. I understand that using a hydrograph with a high flood peak would lead to significantly bigger HR and the consequent loss of life. However, is it not the scope of this model to represent worst-case scenarios to improve flood risk management and reduce loss of people? Also, not necessarily using a shorter hydrograph can lead to loss of life. I invite the authors to run different scenarios keeping the same volume of the input hydrograph but changing the timing of the peak. The timing of the peak is a crucial factor in any flood risk management application and I do not understand why its influence was not included in this study."*

Although not reported in this paper, we previously explored four different hydrographs, as suggested, based on keeping the same volume but changing the timing of the peak (Shirvani et al. 2021). Here, we chose the most suited hydrograph (timing and peak) that generates a full range of flood risk HR states to conduct the aforementioned evaluation for the new version of the flood-pedestrian simulator. Supported by the HR analysis in Shirvani et al.( 2021), this choice for the hydrograph is representative of flood types that frequently occur in urban areas, hence for evaluating the simulator with a focus on individual-scale risk analysis while considering their two-way interaction with the floodwater flow and realistic human-body characteristics. As we already clarified in the initially submitted preprint (see the 3rd paragraph in Sect. 2.3.1), choosing higher peak or shorter timing would lead to bigger HR values associated with loss of life or injury, hence are not suited to test the relevance of this simulator.

> In the revised version of the paper, clarification on the choice of hydrograph has been provided in the 3rd paragraph of Sect. 2.3.1 to clarify:
>
> The flooding inflow was generated based on an inflow hydrograph of a discharge, $Q$ (m$^3$/s) propagating over a duration of 7.5 min, and peaking to 160 m$^3$/s at 3.75 min (Fig. 3). The hydrograph was produced based on the Norwich inundation case study, and because it results in a range for the HR that is inclusive of all the ranges listed in Table 1, i.e. HR < 7 (Shirvani et al., 2021). Deploying a hydrograph with shorter duration or a bigger peak would lead to significantly bigger HR, which is indicative of potential loss of life or injury where a person can take very limited actions to carry on moving to the emergency exit (hence it is outside the scope of this study). When the floodwater starts to propagate over the walkable area, simulation time ($t$) of 0 min, the pedestrian agents start the evacuation and the simulation terminates when all the pedestrian agents have evacuated the walkable area.

- *" Section 2 is a very large and dense section. There are many headings and sub-headings and I found myself lost with a need to scroll up and down. Would not be better to move 2.3.1 and 2.3.2 in a new section 3 called synthetic case study used to test the model and then introduce section 4 (now section 3) on the real-life experiment?"*

Section 2 is focused on the simulator, with Sect. 2.3 dedicated to assessing the relevance of the newly added agent rules and characteristics. As part of Sect. 2.3, the synthetic case study (Sect. 2.3.1 and Sect. 2.3.2) is used to serve the aim of assessing the relevance of these new rules and characteristics within the simulator. Therefore, it makes sense to keep it with the section describing the simulator and its new rules and characteristics.

---

## Author Comment (AC3)

Dear Editor and Referees,

We thank you again for the time spent to evaluate and review our manuscript. This letter includes the final authors' reply to the comments raised by all the three referees and highlights the changes made to manuscript to address the referees' comments.

Therefore, updated versions of our "Reply on RC1" and "Reply on RC2" are included below in this letter, in addition to our "Reply on RC3".

The revised manuscript is also ready to be uploaded in which the changes made are flagged in the red colour to facilitate the re-review process.

Sincerely,
G. Kesserwani and M. Shirvani
* * *
Dear Referee #1,

We thank you for your critical and well-considered comments, which we found helpful to substantially improve the quality of our paper. Therefore, the major and minor comments brought by Referee #1 were all addressed in a revised manuscript.

Below, we will repeat each comment (italic font) and reply directly below it (standard font). After each response, the associated changes applied to the revised manuscript (coloured in red) are included in a table box.

Sincerely,
G. Kesserwani and M. Shirvani

**General comments**
*"The value of Agent-Based Modelling as a technique to understand the role of individual decision making in response to hazards, such as flooding, has increased in recent years, since the initial publication of ABM evacuation behaviour was published in 2011. This study is an interesting example of further development of this application of ABM and demonstrates a good synthesis of empirical research to formulate agent behaviour. Through introducing agent characteristics such as age, gender, weight, variable movement speeds and stability patterns, the model introduces an enhanced level of heterogeneity amongst the agent population from previous iterations of the model."*

**Major comments**
*"However, there lies two fundamental concerns related to the outputs of the model:"*

The two fundamental concerns were insightful for us to carry out the revision work necessary to ascertain the credibility of the model outputs based on which the results were analysed. The minor/technical comments raised were also found valuable to sharpen the quality of the manuscript. A point-by-point reply to each comment is provided below it, together with an explanation of the associated changes made to the manuscript.

*"1. It would appear that uncertainty inherent within modelling stochastic agent behaviour is not acknowledged. The impression given is that model simulations were ran only once for each scenario (lines 331 – 333) and outputs were then analysed. There should be evidence*

*of multiple simulations for each test scenario to account for variance in behaviour of agents. For example, confidence intervals and averaging over an appropriate number of simulations would provide a more representative set of outputs that account for stochasticity."*

The reviewer is correct about this important point. As the outputs analysed in the initially submitted preprint were extracted from a single run, they do not account for the uncertainty inherent within stochastic agent behaviour. Therefore, the preprint has been revised to: first, acknowledge this uncertainty; then, to explore the sample size needed for a series of probabilistic runs to reach statistically significant outcomes for a range of confidence levels; and, finally, reproduce new outcomes averaged out from the probabilistic runs for the two case studies explored and each of their test scenario.

Section 3.2 was therefore added to elaborate on the probabilistic simulation runs including a sensitivity analysis:

**3.2 Simulation runs**

The simulator was executed at a resolution of 2.59 m × 2.59 m for each of the grids of navigation and flood agents. The time-step was taken to be the minimum between the adaptive time-step of the hydrodynamic model and the 1.0 s time-step of the pedestrian model (a visualisation of a simulation run can be found in the video supplement in Shirvani (2021)). In each run, the simulator is set to record the information stored in the flood agents and the pedestrian agents at each time-step. Recorded outputs from a simulation run include the positions of the pedestrian agents, their flood risk states (HR-related) and or their stability states (including toppling-only, toppling-and-sliding and sliding-only conditions[1]). As the motion of each pedestrian agent is governed by a stochastic (space-time) process, series of 10 and 20 simulation runs were conducted to average out a plausible outcome for each of the configuration Modes. The plausibility of the average outputs from both series of runs is evaluated, by estimating the margin of error (*MOE*) assuming confidence levels ranging between 90% and 99.9%. The following formula is used to evaluate the *MOE*:

$$MOE = Z_{score} \times \sqrt{\frac{\sigma^2}{n}}, \tag{3}$$

where, $Z_{score}$ is the critical value, which is equal to 1.65, 1.96, 2.17, 2.58 and 3.29, for confidence levels of 90 %, 95 %, 97 %, 99 % and 99.9 %, respectively (Hazra, 2017); $\sigma$ is the standard deviation from the sample of outputs of size $n = \{10, 20\}$; and $\sigma = \sqrt{\frac{\Sigma(x_i - \bar{x})^2}{n}}$, with $x_i$ representing the number of pedestrians with a particular HR-related flood risk or stability state extracted from the recorded outputs, and $\bar{x}$ is the averaged value. Table 5 lists the maximum *MOE* evaluated for the different confidence levels, with respect to the average number of pedestrian agents under different HR-related flood risk and stability states for configuration Mode 0 to Mode 4.

**Table 5:** Maximum margin of error (*MOE*) for the average number of pedestrian agents with different HR-related flood risk or stability states that are extracted from the recorded outputs of all the configuration modes (Table 4) and across different confidence levels ranging from 90 % to 99.9%. Different ranges of the evaluated maximum *MOE* are highlighted with different colour shades: green, orange and red to indicate *MOE* ≤ ± 5, 6 ≤ *MOE* ≤ 9 and *MOE* ≥ 10, respectively.

| Mode | HR-related flood risk and stability states | Maximum *MOE* | | | | | | | | | |
|---|---|---|---|---|---|---|---|---|---|---|---|
| | | *n* = 10 | | | | | *n* = 20 | | | | |
| | | 90 % | 95 % | 97 % | 99 % | 99.9 % | 90 % | 95 % | 97 % | 99 % | 99.9 % |
| 0 | HR < 0.75 | ± 5 | ± 6 | ± 6 | ± 8 | ± 10 | ± 3 | ± 4 | ± 4 | ± 5 | ± 6 |
| | 0.75 < HR <1.5 | ± 4 | ± 5 | ± 6 | ± 7 | ± 9 | ± 3 | ± 3 | ± 4 | ± 4 | ± 5 |
| | 1.5 < HR <2.5 | ± 3 | ± 3 | ± 4 | ± 5 | ± 6 | ± 2 | ± 3 | ± 3 | ± 4 | ± 5 |
| | HR > 2.5 | ± 1 | ± 1 | ± 1 | ± 2 | ± 2 | ± 1 | ± 1 | ± 1 | ± 1 | ± 1 |
| | Toppling-only | ± 5 | ± 6 | ± 7 | ± 8 | ± 10 | ± 4 | ± 4 | ± 5 | ± 6 | ± 8 |
| | Toppling-and-sliding | ± 4 | ± 5 | ± 5 | ± 6 | ± 8 | ± 3 | ± 4 | ± 4 | ± 5 | ± 9 |
| 1 | HR < 0.75 | ± 6 | ± 7 | ± 8 | ± 9 | ± 12 | ± 4 | ± 4 | ± 5 | ± 6 | ± 7 |
| | 0.75 < HR <1.5 | ± 6 | ± 7 | ± 8 | ± 10 | ± 12 | ± 4 | ± 4 | ± 5 | ± 6 | ± 7 |
| | 1.5 < HR <2.5 | ± 1 | ± 1 | ± 1 | ± 1 | ± 1 | ± 1 | ± 1 | ± 1 | ± 1 | ± 1 |

| | | | | | | | | | | | |
|---|---|---|---|---|---|---|---|---|---|---|---|
| | HR > 2.5 | ± 0 | ± 0 | ± 0 | ± 0 | ± 1 | ± 0 | ± 0 | ± 0 | ± 0 | ± 0 |
| | Toppling-only | ± 6 | ± 8 | ± 8 | ± 10 | ± 13 | ± 4 | ± 4 | ± 5 | ± 6 | ± 7 |
| | Toppling-and-sliding | ± 5 | ± 6 | ± 7 | ± 8 | ± 10 | ± 3 | ± 4 | ± 5 | ± 5 | ± 7 |
| **2** | HR < 0.75 | ± 6 | ± 7 | ± 7 | ± 9 | ± 11 | ± 4 | ± 5 | ± 5 | ± 6 | ± 8 |
| | 0.75 < HR <1.5 | ± 7 | ± 8 | ± 9 | ± 10 | ± 13 | ± 5 | ± 6 | ± 6 | ± 7 | ± 9 |
| | 1.5 < HR <2.5 | ± 1 | ± 1 | ± 1 | ± 2 | ± 2 | ± 1 | ± 1 | ± 1 | ± 1 | ± 1 |
| | HR > 2.5 | ± 1 | ± 1 | ± 1 | ± 1 | ± 1 | ± 1 | ± 1 | ± 1 | ± 1 | ± 1 |
| | Toppling-only | ± 6 | ± 7 | ± 8 | ± 9 | ± 12 | ± 4 | ± 4 | ± 5 | ± 6 | ± 7 |
| | Toppling-and-sliding | ± 6 | ± 7 | ± 7 | ± 9 | ± 11 | ± 4 | ± 5 | ± 5 | ± 6 | ± 8 |
| **3** | HR < 0.75 | ± 6 | ± 7 | ± 8 | ± 9 | ± 12 | ± 4 | ± 4 | ± 5 | ± 6 | ± 7 |
| | 0.75 < HR <1.5 | ± 6 | ± 7 | ± 8 | ± 9 | ± 12 | ± 4 | ± 5 | ± 5 | ± 6 | ± 8 |
| | 1.5 < HR <2.5 | ± 1 | ± 1 | ± 1 | ± 1 | ± 1 | ± 0 | ± 1 | ± 1 | ± 1 | ± 1 |
| | HR > 2.5 | ± 0 | ± 0 | ± 0 | ± 0 | ± 0 | ± 0 | ± 0 | ± 0 | ± 0 | ± 0 |
| | Toppling-only | ± 6 | ± 7 | ± 8 | ± 9 | ± 12 | ± 4 | ± 5 | ± 5 | ± 6 | ± 8 |
| | Toppling-and-sliding | ± 6 | ± 7 | ± 8 | ± 9 | ± 12 | ± 4 | ± 4 | ± 5 | ± 6 | ± 7 |
| **4** | HR < 0.75 | ± 5 | ± 6 | ± 7 | ± 9 | ± 11 | ± 4 | ± 4 | ± 5 | ± 6 | ± 7 |
| | 0.75 < HR <1.5 | ± 7 | ± 9 | ± 10 | ± 12 | ± 15 | ± 5 | ± 6 | ± 6 | ± 7 | ± 10 |
| | 1.5 < HR <2.5 | ± 1 | ± 1 | ± 1 | ± 1 | ± 2 | ± 1 | ± 1 | ± 1 | ± 1 | ± 1 |
| | HR > 2.5 | ± 1 | ± 1 | ± 1 | ± 1 | ± 1 | ± 0 | ± 0 | ± 1 | ± 1 | ± 1 |
| | Toppling-only | ± 6 | ± 7 | ± 8 | ± 9 | ± 12 | ± 4 | ± 4 | ± 5 | ± 6 | ± 7 |
| | Toppling-and-sliding | ± 6 | ± 7 | ± 8 | ± 9 | ± 12 | ± 4 | ± 5 | ± 5 | ± 6 | ± 7 |

For $n = 10$, there is a considerable increase in the maximum *MOE* with Mode 1 to Mode 4 compared to Mode 0. This is particularly seen for the number of pedestrian agents in low and medium flood risk states (HR < 0.75 and 0.75 < HR <1.5, respectively) and with toppling-only and toppling-and-sliding stability states. This suggests that the more sophisticated the pedestrian agent characteristics and rules, more discrepancies would appear in the simulator's outcomes. The maximum *MOE* identified suggests a deviation of around ± 15 from the averaged outcomes. However, when the sample size is increased to $n = 20$, the maximum margin of error does not exceed ± 10 for all the modes and confidence levels. Therefore, the simulation results analysed next are averaged out from a sample of 20 simulation runs, subject to ± 10 maximum *MOE* for a population of 1000 pedestrians in the flooded walkable area, which corresponds to a variance of 1 %.

With the changes above, Sect. 3.3 has been revised too in order to incorporate new figures and accommodate their respective changes in the text.

Section 4.2.3 has also been revised to confirm the appropriateness of using a series of 20 simulation runs to average out the outputs for the real-case study:

**4.2.3 Simulation runs**
A series of 20 simulation runs was performed under configuration Mode 2 for each of the 20 %, 30 % and 40 % threshold for the 'autonomous change of direction' condition (visualisation of a simulation can be found in the video supplement in Shirvani (2021)). Each run was set to start at $t = - 10$ min to allow the floodwater to propagate during 10 min so that the evacuation process starts at $t = 0$ min. Outputs averaged from each series of simulation included spatial and temporal information, at each time step, about the pedestrian agents as they evacuate ($t > 0$ min). The averaged outputs include the position, HR-related flood risk state, stability state (with a toppling-only condition, toppling-and-sliding condition and sliding-only condition), and the choice for the destination selected by the pedestrian agents during the evacuation process. Considering the stochastic uncertainties associated with the motion of the pedestrian agents, the plausibility of the averaged outputs from the 20 runs was evaluated. The evaluation was based on the *MOE*, using Eq. (3), for 99.9 %

confidence level only, informed by the results of the analysis in Sec. 3.2. Table 6 shows the maximum *MOE*s found for the number of pedestrians predicted to be in the considered HR-related flood risk and the stability states, obtained from the 20 runs using each of the 20 %, 30 % and 40 % threshold, respectively. It can be seen that the maximum *MOE* increases as the risk perception level decreases, suggesting a notable increase in the uncertainty after the incorporation of the risk perception component into the modelling of pedestrian behaviours.

**Table 6:** Maximum margin of error (*MOE*) for the average number of pedestrian agents with different HR-related flood risk or stability states that are extracted from the recorded outputs throughout the simulations for each 20 %, 30 % and 40 % threshold. Different ranges of the evaluated maximum *MOE* are highlighted with different colour shades: green, orange and red to indicate $MOE \leq \pm 5$, $6 \leq MOE \leq 9$ and $MOE \geq 10$, respectively.

| HR-related flood risk and stability states | Maximum *MOE* | | |
|---|---|---|---|
| | 20 % threshold | 30 % threshold | 40 % threshold |
| HR < 0.75 | ± 16 | ± 16 | ± 19 |
| 0.75 < HR <1.5 | ± 2 | ± 8 | ± 15 |
| HR > 1.5 | ± 0 | ± 1 | ± 2 |
| Toppling-only | ± 2 | ± 5 | ± 13 |
| Toppling-and-sliding | ± 1 | ± 4 | ± 7 |

Next, the averaged outputs are analysed for each of the 20 %, 30 % and 40 % thresholds, considering the popularity of the destination selected by the pedestrian agents (among south, east and north) together with their HR-related flood risk and stability states.

With the changes above, Sect. 4.3 has been revised too in order to incorporate new figures and accommodate their respective changes in the text.

*"2. The number of agents included in the Hillsborough case study is 4,080. The lowest recorded attendance for Sheffield Wednesday in 2019 was counted at 21,485, meaning that the agent population is only 18% of what is considered 'real'. Given that the start of the paper states that the FLAMEGPU platform can handle "as large population size as needed (line 79)" and that throughout the paper congestion is frequently stated as a factor that influences flow dynamics and evacuation time, a question emerges as to why such a significantly low agent population has been adopted. The paper states that "using a bigger population size would lead to extreme pedestrian congestion that impacts the movements of individual pedestrians (lines 489 – 490)".Surely, in the context of a football match, this is an important factor that must be represented as accurately as possible. There is no justification for the chosen figure of 4,080 and it seems arbitrary. I am not wholly convinced that subsequent outputs reflect evacuation times that would be representative of a football match. Either a more realistic agent population in line with actual attendance rates is necessary, or justification that the emergent behaviour is not dramatically impacted by the chosen number of spectators."*

The reviewer raised fair points regarding the lack of justification of the selected 4,080 agent population and of how effects of pedestrian congestion on the evacuation times were handled. In terms of agent population, the average recorded attendance should be around 24,000 in normal weather conditions. As the case study assumes severe weather conditions and flood warnings were issued before the event, it makes sense to consider around 20 % of the expected spectators. This is also representative of the percentage of people who would ignore the warnings and attend the match (Fielding et al., 2007).

As the statement: *"using a bigger population size would lead to extreme pedestrian congestion that impacts the movements of individual pedestrians"* was a major oversight, it has been removed from the manuscript. Instead, new clarifications are provided on how a dischach rate was implemented into the simulator for this case study to comply with a safe evacuation measure, in other words ensure that *"the emergent behaviour is not dramatically impacted by the chosen number of spectators"*.

> End of Sect. 4.1 has been extended to more clearly justify the reduced number of spectator selected and to also justify the outcomes of the simulator won't be significantly impacted for a higher population size.
* * *
…

To do so, it was assumed that the site in Fig. 7 is hit by a flood during a football match where the spectators are caught unaware of the rainfall accumulation around the stadium, similar to the event that could have happened in 2019. As discussed before, the floodwater is likely to accumulate from the north and east sides to move downhill towards the main entrance of the stadium. Once the floodwater has reached the stadium's main entrances, an emergency evacuation alarm is issued, urging people to start evacuating immediately. The spectators are then put into queues inside the stadium to be evacuated towards the walkable area. The evacuating spectators gradually enter the walkable area where they get in direct contact with flooded areas along their ways to any of the south, east or north destinations. In this scenario, a population of 4,080 spectators was assumed, which is lower than normal due to the severe weather condition and flood warnings issued prior to the event. This population is around 20 % of the spectators expected, and represents the relative number of people who would ignore the warnings and attend the match (Fielding et al., 2007).

For this case study, a dispatch measure was introduced to the simulator to release the evacuees into the walkable area during the flooding. The dispatch measure limits the influx rate to person-per-second per width unit to comply with guidance methods for controlling the density of large crowds outside the stadiums for safe evacuation (Minegishi and Takeichi, 2018; Still, 2019). For a gate that is around 4 m wide, four pedestrians per second are dispatched from the stadium to the walkable area. Using the simulator with this dispatch rate limits the overall number of pedestrians that would be present in the walkable area at a time. Therefore, running the simulator to analyse the evacuation of a larger number of spectators is expected to lead to similar risk trends based on pedestrians' different HR-related flood risk and stability states, which would only be prolonged over a larger evacuation time.

The flood-pedestrian simulator is applied to analyse how the number of pedestrians with different HR-related flood risk and stability states change under this scenario with a further focus on modelling their preference for the destination choice during the flood evacuation, by activating the 'autonomous change of direction' condition (Sect. 2.2.3).

**Technical/Minor Comments:**

- *"In section 2.3.2, it states "…the more crowding of pedestrians the more energy loss in the floodwater dynamics for low risk floodwaters, which in turn enables the pedestrians located behind to take a faster moving speed (lines 346 – 348)". Whilst it is well noted that crowding would disperse the floodwater, making pedestrian movement easier, I can't help but wonder if collective moving speed would in turn be slowed down by the congestion itself, therefore this could be acknowledged."*

While revising Sect. 3.3 (or Sect. 2.3.2 in the initially submitted preprint), the above-raised aspect has been addressed to acknowledge the influence of pedestrian congestions in the floodwater on the collective moving speed. Our findings suggest that, with the two-way interaction condition, crowding of pedestrians disperse the floodwater, in turn allowing more spreading of the moving pedestrians and those in the front to move faster (see revised Fig. 6 below). However, the collective moving speed does not exhibit any significant difference due to congestion, as can also be noted in the overall trends of HR-related flood risk states of pedestrians (addressed in the discussion of the revised Fig. 4 in Sect. 3.3).
* * *
> The discussion of Fig. 4a in the 1st paragraph of Sect. 3.3 has been revised to acknowledge this aspect:
* * *
**3.3 Analysis of flood risks to people**

Figure 4 shows the trends in the number of evacuating pedestrians with different HR-related flood risk states predicted by the simulator after 20 runs using all the configuration modes (Table 4). Figure 4a represents how the number of pedestrians with a low flood risk state (HR < 0.75) change during 20 minutes of flood time. Figure 4a-left includes the trends predicted after enabling the walking condition for the age-related moving speeds (Mode 1) versus those predicted by further enabling the two-way interaction condition (Mode 2). In Mode 1, the trend is in good agreement with the

baseline predictions (Mode 0, with non-age related moving speeds) at flooding times when there are less than 100 pedestrians in the walkable area with a low flood risk state, during 3.5 min to 7 min. A considerable difference among the predictions starts to appear when more than 150 pedestrians are present, around 2.5 min and 8.5 min. This difference seems to impact the overall trend, suggesting a 6 min longer duration with a higher number of pedestrians being predicted to be under this flood risk state, during 8 min to 18 min. In Mode 2, compared to Mode 1, the number of evacuating pedestrians is seen to reduce further at flooding times involving more than 150 pedestrians, around 2.5 min and 10 min. This is expected as crowding of pedestrians in low risk floodwaters would disperse the floodwater dynamics, which in turn help pedestrians evacuating ahead to pick up a faster moving speed (Shirvani et al., 2021). This does not seem to influence the collective moving speed of pedestrians, for example by generating additional congestions (as shown later in Fig. 6), as the overall trends with Mode 1 and Mode 2 are very close. Figure 4a-right contrasts the trends predicted after activating the running condition for the age-related moving speeds (Mode 3) to those predicted by also enabling the two-way interaction condition (Mode 4). In Mode 3 and Mode 4, the trends show a considerably faster moving speed of pedestrians (than with Mode 1 and Mode 2), significantly reducing the duration when pedestrians fall under a low risk state, suggesting outputs that are close to the baseline predictions (Mode 0). With Mode 3, discrepancies (compared with Mode 0) only occur between 2.5 and 3.5 min and after 8 min of flooding, when there are more than 150 pedestrians moving under the running condition. In Mode 4, with further enabling the two-way interaction condition, the trends remain close to those predicted under Mode 3, except at 2.5 min flooding time that involves more than 200 pedestrians under a low flood risk state. This suggests that activating the two-way interaction condition with the running condition may only temporarily influence the pedestrians' collective moving speed, namely when more than 200 pedestrians are caught under a low flood risk state. Overall, there is a major difference in the collective moving speeds of pedestrians when age-related walking vs. running speeds are deployed, leading to prolonged vs. shortened evacuation times compared to the baseline predictions (Mode 0), respectively. Also, using the two-way interaction condition seems to be a sensible choice for simulating mass pedestrian evacuations in low risk floodwater.

- *"In section 2.3.2, it also states "enabling it with the two-way condition (Mode 2) increases slightly the evacuation time as crowding is more likely under the walking condition (lines 435 – 436)". It would be reasonable to assume that the 'running condition' would be more likely to cause crowding as it represents a more 'excited' response, causing further bottlenecking in the evacuation process compared to more organised walking agents. Some consideration of this would be insightful."*

As shown in the revised Fig. 6, the reasonable assumption raised above is retrieved by the simulator when running with the two-way interaction condition and contrasting pedestrian distribution at 6 min when largest population of pedestrians are at risk of toppling and the medium risk floodwater (0.75 < HR < 1.5) affect the majority of the population.

New Fig. 6 has been produced and discussed in Sect. 3.3 to include insights on this aspect:

…

Figure 6 compares the spatial distributions of the evacuating pedestrians over flood HR map at flood time 6 min, obtained from simulator runs under Mode 1 to Mode 4. In each of the sub-plots, the framed $50 \times 50$ m$^2$ before the emergency exit includes the number of pedestrians in that area, where the congestion of pedestrians is assessed for the different modes. With all the modes, the simulator predicted a dominance of medium risk floodwaters (0.75 < HR < 1.5) over the walkable area, causing the majority of the pedestrians to fall into a toppling-only condition (purple dots) and a minority to have a stable condition (green dots) in front of the emergency exit and from the left side of the crowd. By contrasting the spatial distribution of pedestrians obtained from Mode 1 and Mode 2 (upper panels), there seems to be a considerable increase in the number of pedestrians with a stable condition when the two-way interaction condition is enabled with the walking condition (Mode 2). The same pattern is observed with Mode 3 and Mode 4 (lower panels), but this is accompanied by a shift in the position of pedestrians towards the front, as expected for the running condition. On the other hand, by contrasting the number of pedestrians in the small square obtained from Mode 1 and Mode 3 (left panels), it can be observed that enabling the running condition results in a decrease in the congestion of pedestrians in

front of the emergency exit. The opposite pattern is observed when enabling the two-way interaction condition in Mode 2 and Mode 4, showing an increase in the congestion of pedestrians under a running condition compared to the walking condition. Hence, using the two-way interaction condition with the simulator may be useful to more realistically evaluate bottlenecking impacts of an evacuation process.

[Figure]

**Figure 6:** Spatial distribution of pedestrian agents, represented by coloured dots, predicted by the simulator under Mode 1 to Mode 4 at 6 min after flooding. The grey colour represents the floodwater extent based on the flood HR quantity and the square before the emergency exit represents an area of $50 \times 50$ m² with a number printed alongside it representing the number of pedestrians in that area.

…

- *"The model demonstrates well a fundamental concept inherent within complexity sciences and Agent-Based Modelling; emergence. The adoption of risk perception thresholds seem to have influenced the favouring of destination selection by agents with varying levels of risk perception. As is the case for risk perception and adaptive behaviour in response to flooding (i.e. risk perceptions thresholds increase after an individual's property is flooded, which increases the likelihood of adopting protective measures in future) as an active area of interest within flood risk management. Some further comments on the overlap between risk perception and evacuation behaviour may be insightful to lay foundations for future research."*

In the revised paper, comments have been added to elaborate on the overlap between risk perception and evacuation behaviour, supported by citing and discussing recently published papers that reviewed the impact of past flooding experience of individuals on behaviours. Also, the uncertainty analysis added to Sect. 4.2.3 (addressing major comment #1) seem to offer new insights on the impact of adding the risk perception on the variance in the simulation outcomes.
* * *
Changes made to Sect. 2.2.3:
* * *
**2.2.3 Autonomous change of direction condition**

... The first rule makes a pedestrian agent detect and choose another destination if the floodwater depth along its way becomes higher than a threshold of a floodwater depth to body height. The choice for the threshold is case-dependent and exploring different thresholds may be necessary (Sect. 4.2.2) as an individual's flood risk perception is dependent on different factors, including past flooding experiences (Hamilton et., 2020; Abebe et al., 2020). This affects the modelling of decisions, i.e. when and where people enter the floodwater or make a move into another destination (Becker et al., 2015; Netzel et al., 2021). Applying this rule enables the pedestrian agents to make decisions on which pathway to take within an environment layout where there is no specific emergency exit at time of evacuation. ...
* * *
Also, the following changes are made in the 3rd paragraph of Sect. 4.2.2:
* * *
…
        For the 'autonomous change of direction' condition, three thresholds of floodwater depth to the body height (Fig. 11) were selected, informed by the experiments in Dias et al. (2021). This was done to account for the uncertainty associated with individuals' different risk perception. The '20 % threshold' was defined to represent people with high-risk perception, such as those who previously experienced a critical flooding incident, and decide not to enter floodwater with a depth that is more than 20 % of their body height. This threshold is estimated based on the ratio of the dominant minimum value for the depth of floodwater that can occur over the walkable area (0.3 m) to the height of the shortest pedestrian agent available (1.4 m). With this threshold, the likelihood of the entire population to be in a condition to change their direction is ensured. The '40 % threshold' was defined to represent people with low-risk perception, such as those who have not yet experienced a flood incident, and decide to enter a floodwater with a depth that is even more than 40 % of their body height. This threshold is estimated based on the ratio of the dominant maximum depth of floodwater (0.9 m) to the height of the tallest population of pedestrian agents available (2.1 m). This threshold enables the entire population to have the freedom to keep moving even within the deepest floodwater in the walkable area (0.9 m). The '30 % threshold' accounts for an average-risk perception, such as those who previously experienced a minor to moderate flooding incident. Pedestrians with average-risk perception would decide to enter floodwater up to their knees, which constitutes 30 % of the human body height (Teichtahl et al., 2012).

[Figure]

**Figure 11:** Thresholds of floodwater depth to body height that are specified for pedestrian agent to accommodate uncertainty associated with different risk perception of people in the real-world case study

Recall also the above-reported revisions applied to Sect. 4.2.3, addressing major comment #1, which demonstrates the impact of the risk perception on the variability in the simulator outcomes.

■ *"Some more clarity on what solvers are used to dictate the flow of water in the hydrodynamic model should be provided in Section 3.2.1."*

Clarification has been provided in the 2nd paragraph of Sect. 2.1 instead, as it is the most appropriate location to include this general information about the simulator's make-up.

Changes made to Sect. 2.1:

...The environment layout encodes force vector fields providing navigation to key destinations. These fields are stored within a grid of fixed discrete agents, forming a navigation map (Karmakharm et al., 2010). The navigation map is necessary for pedestrians' way-finding decisions while they are directed to reach their key destinations.

   The hydrodynamic model is formulated based on a non-sequential implementation of a finite volume solver of the depth-averaged shallow water equations on a two-dimensional grid on FLAMEGPU, which was validated previously in Shirvani et al. (2021). The hydrodynamic model was applied on another fixed grid of discrete agents, flood agents, which is coincident with the grid of navigation agents...

■ *"Calibration/validation evidence of the hydrodynamic model should be provided in more detail to highlight the robustness of hydrodynamic outputs. Two separate figures are provided of flood extents; an Environment Agency figure (Figure 8) and your own (Figure 10). These could be combined into one figure, by overlaying and analysing to provide an FSTAT value to indicate accurate representation of simulated outputs versus simulated. Similarly (at a minimum), a figure showing simulated and observed hydrographs should be provided to demonstrate that flood depths are within an acceptable tolerance."*

The revised Sect. 4.1 and Sect. 4.2 provide detailed clarification that the Environment Agency (EA) maps were used as evidence to highlight the importance of the selected site for the purpose of the study, not as reference to validating the robustness of the hydrodynamic outputs. The EA maps can only be useful to ensure that the hydrodynamic model predicts ranges of floodwater depth and velocity magnitude that are within the expected ranges. To the best of our knowledge, there is no record of any observed hydrograph sampled at a gauge point located in the selected study site. Therefore, the simulated time series for the mean water depth and velocity magnitude were added to Fig. 10, to show that the ranges of the floodwater depth and velocity would fairly agree with the approximate ranges reported by the EA when the evacuation starts.

Changes made to the 3rd paragraph of Sect. 4.1:

...

This site, being both adjacent to River Don and located down the hills where rainwater runoff accumulates, has been flagged to be prone to future pluvial or fluvial flood types according to the EA's flood information service that is available online at https://flood-warning-information.service.gov.uk/long-term-flood-risk. This service provides flood maps for identifying long-term risks in parts of the UK towns based on a 'low', 'medium' and 'high' annual probability of occurrence. By entering the Hillsborough stadium postcode, S6 1SW, the flood maps showing the approximate ranges of the expected floodwater depth and velocity magnitude for the study site (Fig. 7) were obtained, as shown in the screenshots in Fig. 8. The floodwater depth map associated with a high annual probability (left panel, Fig. 8a) represents the least extreme scenario, where the range for the floodwater depth is likely to vary between 0.3 m and 0.9 m to potentially cover the northern branch of the walkable area with velocity magnitudes greater than 0.25 m/s. For a medium annual probability of occurrence (middle panel, Fig. 8a), the flooding extent could widen to potentially obstruct both northern and eastern branches with the range of floodwater depths reaching beyond 0.9 m and much wider extent for velocity magnitudes greater than 0.25 m/s mostly along the eastern branch (middle panel, Fig. 8b). For a low annual probability of occurrence (right panel, Fig. 8a), an even wider flood extent would be expected up to almost submerging the entire walkable area with dominance of deeper than 0.9 m floodwater depths along the northern branch and higher than 0.25 m/s velocities at the north, east and the sides of the southern branch. Even in the most optimistic flooding scenario, at least the northern branch near the stadium's entrance would be affected, where an evacuating spectator during a flood has to wade through floodwaters at a depth that is between 0.3 m and 0.9 m and velocities higher than 0.25 m/s. Therefore, investigating the dynamics of how people respond in a during-flood evacuation is of paramount importance for the selected study site.

[Figure]

**(a)**

● $v$ greater than 0.25 m/s    ● $v$ less than 0.25 m/s    ◣ Direction of water flow

**(b)**

**Figure 8:** Screenshots of EA's flood risk maps of the study site showing the extent of flooding from surface water with 'low', 'medium' and 'high' annual flooding probabilities featuring different floodwater ranges of: (a) depth and (b) velocity. These screenshots were retrieved from https://flood-warning-information.service.gov.uk/long-term-flood-risk (credit: © Crown and database rights under Open Government Licence v3.0).

Changes made to the 3rd paragraph of Sect. 4.2:

**4.2.1 Hydrodynamic model set-up**

The hydrodynamic model was set up to run on a grid of 128 × 128 flood agents. The grid of flood agents (equally for the grid of navigation agents) was set to store the terrain features of the study site, loaded from a digital elevation model (DEM) at 1 m resolution, which is available online from the UK's Department for Environment Food & Rural Affairs (DEFRA) LiDAR Survey at: https://environment.data.gov.uk. To the best of the authors' knowledge, there is no record of any observed hydrograph sampled at a gauge point located in the selected study site. Therefore, the flooding flow was generated by formulating an inflow hydrograph based on the November 2019's rainfall volume (Fig. 9). The hydrograph was set to replicate a total runoff volume accumulation of 1,045.3 m³ based on a 0.0638 m rainfall over the entire 16,384 m² site. This volume was estimated using the direct runoff method: *rainfall volume* (m³) = *rainfall height* (m) × *area* (m²). The hydrograph was generated as:

$$Q_t = Q_{initial} + (Q_{peak} - Q_{initial})(\frac{t}{t_{peak}}.exp(\frac{1-t}{t_{peak}}))^{\beta}, \tag{4}$$

where $Q_t$ (m³/s) is the inflow discharge propagating along the north-east boundary intersecting the eastern branch; $Q_{peak}$ (m²/s) = 0.29 is the peak discharge, that was calculated by distributing the runoff volume (1,045.3 m³) per second over an hour of flooding; $Q_{initial}$ (m³/s) is the initial discharge, taken 0 m³/s; $t$ (min) is the simulation time varying between 0 to 10 min; $\beta$ = 10 is a constant to soften the shape of the hydrograph and $t_{peak}$ (min) = 5 is the time of peak discharge. This choice, for $t_{peak}$, considers the peak discharge has been reached halfway during the flooding to cause the propagating floodwater to reach to the main stadium's entrances by 10 min leading to triggering the evacuation alarm.

[Figure]

**Figure 9:** Inflow hydrograph produced by Eq. (4) used to generate the floodwater propagation occurring from the north-east side of the site

To ensure that the resulting ranges of floodwater depth and velocity magnitude generated by the hydrograph in Fig. 9 fit the expected ranges of floodwater depth and velocity reported by the EA, a run was conducted without pedestrian consideration. Fig. 10a shows the map of the predicted floodwater depth after 10 min of flooding, while Fig. 10b and Fig. 10c includes the time series of the mean floodwater depth ($\overline{h}$) and velocity magnitude ($\overline{V}$) in the lead-up to 10 min, respectively. From the floodwater depth map, it can be seen that the spatial distribution of floodwater depth varies between 0.3 m and 0.9 m inside the walkable area at the time when pedestrians start to evacuate. By this time, Fig. 10b

and Fig. 10c suggest that the mean floodwater depth is at its deepest level of 0.5 m and the velocity magnitude reduces to 1.5 m/s. Beside confirming that the generated hydrograph leads to a realistic flood event in line with the EA's expectations, these results indicate that a pedestrian evacuating into the floodwaters shown in Fig. 10a would be under a low-to-medium flood risk state with an HR value estimated around 1 (can be extracted by the end of the time series in Fig. 10b and Fig. 10c).

[Figure]

**(a)**                    **(b)**                    **(b)**

**Figure 10:** Outputs of the simulator generated after and during 10 min of a single hydrodynamic run without pedestrian consideration plotted in terms of: (a) floodwater depth map and temporal changes in the average floodwater in terms of (b) depth and (c) velocity.

■    *"A supplementary table providing a summary of all agent variables and values upon initialisation (flooding, navigation and pedestrian agents) would be useful. This would promote replicability and reproducibility."*

The *'Flood-pedestrian simulator user guide'* file on Zenodo (10.5281/zenodo.4672125) includes a summary of all agent variables and values upon initialisation, and many other informative steps to promote replicability and reproducibility.
* * *
Dear Referee #2,

We thank you for the useful review comments and discussions. Answers to the issues raised by Referee #2 are provided below, and have been addressed in a revised manuscript.

Below, we will repeat each comment (italic font) and reply directly below it (standard font). After each response, the associated changes applied to the revised manuscript (coloured in red) are included in a table box.

Sincerely,
G. Kesserwani and M. Shirvani

*" The paper "Flood-pedestrian simulator for modelling human response dynamics during flood-induced evacuation: Hillsborough stadium case study" aims at showing the effect of human-body characteristics and in-model behavioural rules when included in an ABM integrated framework for flood evacuation modelling. The model is largely based on the previous version developed by the authors, and it is tested on a synthetic case study and on a real-world case. The topic of this study is definitely highly relevant and timely. However, I think that the study could be further strengthened by assessing the effect of the evacuation time and characteristics of the inflow hydrograph on the evacuation process. These factors could have a higher impact than the human-body characteristics considered by the authors for flood risk mitigation purposes."*

Thank you for highlighting the timeliness and relevance of the topic investigated. Before we respond to the comments, we clarify that the presented version of the simulator is enhanced with empirically-derived behavioural rules to particularly:

account for the realistic heterogeneity in pedestrian agents' characteristics (human age, gender and body mass) and behaviours (mobility states, risk perception thresholds); and, their back interaction on the floodwater dynamics. In the context of microscopic evacuation modelling in small urban areas, both the heterogeneity and the back interaction factors could play a determinant role to assess during-flood risk to people and the potential changes in their spatial and temporal evacuation patterns. In this context, the scope of this paper has been to explore the impact of pedestrian characteristics and behaviours on evacuation timing and risk assessment, and not the evacuating timing on general risk assessment.
* * *
In the revised introduction (3rd paragraph, Sect. 1), the scope of study has been clarified, while also acknowledging the importance of evacuation time and characteristics of the inflow hydrograph as the reviewer suggested:
* * *
… These ABM-based evacuation simulation tools were developed to inform emergency plans for severe flood types, such as in the immediate aftermath of a dam-break or a tsunami wave (e.g. Lumbroso et al., 2021). The focus of these tools is mainly on estimating the loss of life, pinpointing bottlenecks and high-risk areas, and assessing how flood warnings of an impending flash flood could reduce the number of casualties and injuries. For this type of risk analysis, individuals' microscopic decisions and actions are considered insignificant in influencing the overall simulation outcomes due to the scale and speed of floodwater flow. However, for the most common flood types in urban areas, e.g. surface water due to extreme rainfall, less attention has been given to model the microscopic responses, down to the scale of the moving individuals, in and around flooded urban hubs (Ramsbottom et al. 2006). In this context, Bernardini et al. (2021) imported outputs of a flood model into a commercially available evacuation modelling tool, called MassMotion, to analyse flood risk differences in microscale and macroscale modeling with and without including pedestrians' microscopic evacuation behaviour. They concluded that incorporating pedestrians' microscopic evacuation behaviour in microscale modelling could significantly influence the spatial and temporal changes in flood risk to people, i.e. up to 15 % in absolute terms, when compared to macroscale modelling. Their findings also suggest the need to further incorporate non-homogeneous characteristics of people in a more flexible microscale modelling framework, which may result in additional differences to the analysis of flood risk to people.
* * *
Abstract was also revised as below to make this aspect clear:
* * *
**Abstract.** The flood-pedestrian simulator uses a parallel approach to couple a hydrodynamic model to a pedestrian model in a single agent-based modelling (ABM) framework on Graphics Processing Units (GPU), allowing dynamic exchange and processing of multiple agent information across the two models. The simulator is enhanced with more realistic human-body characteristics and in-model behavioural rules. The new features are implemented in the pedestrian model to factor in age- and gender-related walking speeds for the pedestrians in dry zones around the floodwater and to include a maximum excitement condition. It is also adapted to use age-related moving speeds for pedestrians inside the floodwater, with either a walking condition or a running condition. The walking and running conditions are applicable without and with an existing two-way interaction condition that considers the effects of pedestrian congestion on the floodwater spreading. A new autonomous change of direction condition is proposed to make pedestrian agents autonomous in way-finding decisions driven by their individual perceptions of the flood risk or the dominant choice made by the others. The relevance of the newly added characteristics and rules is demonstrated by applying the augmented simulator to reproduce a synthetic test case of a flood evacuation in a shopping centre, to then contrast its outcomes against the version of the simulator that does not consider age and gender in the agent characteristics. The enhanced simulator is demonstrated for a real-world case study of a mass evacuation from the Hillsborough football stadium, showing usefulness for flood emergency evacuation planning in outdoor spaces where destination choice and individual risk perception have great influence on the simulation outcomes.
* * *
Having clarified that, a point-by-point reply to each comment raised is provided next, with an explanation of the associated changes made to the revised manuscript.

- *" It has been recently shown that the timing in which the evacuation is issued is crucial for reducing flood risk (Alonso et al., 2020). However, in section 2.3.1 the authors state that "When the floodwater starts to propagate over the walkable area, simulation time (t) of 0 min, the pedestrian agents start the evacuation ..." In a no-flooding situation, agents are randomly moving based on their behaviours and on their daily routines. Why an agent should start moving exactly when the water starts entering the building and not before or after? Are there supporting evidence to justify such an assumption? Then, when flooding occurs, there can be two extreme situations. On the one hand, if the agent is doing something else it may not notice the flooding until the pedestrian agent goes on a flooded area. On the other hand, the agent could be informed earlier about a coming flood and start to evacuate earlier. These two scenarios could have dramatic consequences. I suggest including more modes (table 6) accounting for different evaluation timing. This study could show the role of human-body characteristics and in-model behavioural rules in reducing flood risk when evacuation is issued late. "*

To evaluate the relevance of newly-added pedestrian agent's human-body characteristics and in-model behavioural rules, we used Take a Previous model and Add Something (TAPAS) approach, as justified in Sect. 3 (or Sect. 2.3 in the initially submitted preprint). To do that, we had to reconsider the same synthetic test case of the shopping centre (Sect. 3.1 in the revised manuscript) under the same flooding condition investigated before with the previous simulator that involves less sophisticated pedestrian agent characteristics and rules (Shirvani et al., 2020). This test case is designed to evaluate the simulator in capturing (spatially and temporarily) the two-way dynamic interactions between the pedestrians and the floodwater flow. Therefore, the evacuation was set to happen at the onset of flooding to produce the most critical scenario that would cause the maximum interactions between moving pedestrians and flowing floodwater. This assumption is based on the fact that pedestrians are mainly reliant on their own visual detection of the upcoming hazard and/or an immediate announcement in such a small urban hub. In addition, the selected evacuation timing allows us to analyse the relevance of the newly-added characteristics and rules across a full range of flood risk HR states. This means that further exploring pre-flood or post-flood evacuation scenarios, while feasible, won't allow us to assess the potential impacts neither from the added rules and characteristics nor from the during-flood interactions.

- *" My second concern relates to the shape of the hydrograph considered in the synthetic experiment. I understand that using a hydrograph with a high flood peak would lead to significantly bigger HR and the consequent loss of life. However, is it not the scope of this model to represent worst-case scenarios to improve flood risk management and reduce loss of people? Also, not necessarily using a shorter hydrograph can lead to loss of life. I invite the authors to run different scenarios keeping the same volume of the input hydrograph but changing the timing of the peak. The timing of the peak is a crucial factor in any flood risk management application and I do not understand why its influence was not included in this study."*

Although not reported in this paper, we previously explored four different hydrographs, as suggested, based on keeping the same volume but changing the timing of the peak (Shirvani et al. 2021). Here, we chose the most suited hydrograph (timing and peak) that generates a full range of flood risk HR states to conduct the aforementioned evaluation for the new version of the flood-pedestrian simulator. Supported by the HR analysis in Shirvani et al.(2021), this choice for the hydrograph is representative of flood types that frequently occur in urban areas, hence to evaluate the enhanced simulator with a focus on individual-scale risk analysis while considering their two-way interaction with the floodwater flow and realistic human-body characteristics. As we already clarified in the 3$^{rd}$ paragraph of Sect 3.1 (or Sect. 2.3.1 in the initially submitted preprint), choosing higher peak or shorter timing would lead to bigger HR values associated with loss of life or injury, hence are not suited to test the relevance of this simulator.
* * *
In the revised version of the paper, clarification on the choice of hydrograph has been provided in the 3$^{rd}$ paragraph of Sect. 2.3.1 to clarify:
* * *
…
        The flooding inflow was generated based on an inflow hydrograph of a discharge, $Q$ (m$^3$/s) propagating over a duration of 7.5 min, and peaking to 160 m$^3$/s at 3.75 min (Fig. 3). The hydrograph was produced based on the Norwich inundation case study, and because it results in a range for the HR that is inclusive of all the ranges listed in Table 1, i.e. HR < 7 (Shirvani et al., 2021). Deploying a hydrograph with shorter duration or a bigger peak would lead to significantly bigger HR, which is indicative of potential loss of life or injury where a person can take very limited actions to carry on

> moving to the emergency exit (hence it is outside the scope of this study). When the floodwater starts to propagate over the walkable area, simulation time ($t$) of 0 min, the pedestrian agents start the evacuation and the simulation terminates when all the pedestrian agents have evacuated the walkable area.

■ *" Section 2 is a very large and dense section. There are many headings and sub-headings and I found myself lost with a need to scroll up and down. Would not be better to move 2.3.1 and 2.3.2 in a new section 3 called synthetic case study used to test the model and then introduce section 4 (now section 3) on the real-life experiment?"*

As this issue has also been raised by Referee #3, we moved Sects. 2.3.1 and 2.3.2 to a new Sect. 3 dedicated to evaluating the newly added characteristics and rules over the synthetic test case. The real-world case study is now embedded in Sect. 4 as advised by the referee.
* * *
Dear Referee #3,

We thank you for the useful comments that helped us revise the manuscript to particularly stand out its novelty, and improve its organisation and readability.

Below, we will repeat each comment (italic font) and reply directly below it (standard font). After each response, the associated changes applied to the revised manuscript (coloured in red) are included in a table box.

Sincerely,
G. Kesserwani and M. Shirvani

*" It is an important topic and well within the remits of the journal. They have used a single agent-based modelling (ABM) framework running on Graphical Processing Units (GPU). However, in the review's view, further clarification and revision is required before paper is suitable for publication."*

Thank you for highlighting the importance and relevance of the topic and its fit to the NHESS journal. Answers to the issues raised are provided in the following.

■ *" This paper is a follow up from previous papers from the authors and the authors need to clearly state what are the novelties and to demonstrate clearly the effects of these novelties. "*

As acknowledged by Referee # 1, the enhanced simulator incorporates a higher level of heterogeneity for agent population characterisation and employs experimental data to more realistically govern their behavioural rules. Also, the simulator has been featured with new functionalities to make it suited for evacuation planning in outdoor spaces, where pedestrians' autonomous decision making on the selection of the safest destination could be driven by their personal risk perception of the local floodwater (Sect. 2.2.3). The influence of the new features on the simulation outcomes has been demonstrated:
- by first contrasting them against the outcomes from the previous version of the simulator for a synthetic case study of a during-flood evacuation in a shopping centre (Sect. 3); then,
- through a new real-world case study of a mass evacuation from the Hillsborough football stadium during a flood emergency replicating the conditions of November 2019 Sheffield floods.

We have revised the last paragraph of Sect. 1 to ensure that these innovations are clear to the readers.

Changes made to the last paragraph of Sect. 1:

… Also, the latter version of the simulator was only applied to a synthetic test case and it was limited to a simplified way-finding decision rule for directing pedestrian agents to one fixed emergency exit destination (specified in advance). This means that the influence of the interplay between the two-way interaction condition and the pedestrian agent characteristics and rules on the simulation outcomes remained unexplored for real-world scenarios.

This paper presents new developments in the flood-pedestrian simulator for incorporating a higher level of heterogeneity in pedestrian agent characterisation and more realistic behavioural rules than its previous version. The simulator is now augmented for real-world applications with new capabilities to account for:

- age, gender, body height and mass distribution of a subject population;
- age- and gender-related variable moving speeds of individuals in both dry and flooded zones based on real-world datasets and experimental information; and
- autonomous decision making of individuals in choosing one of multiple emergency exit destinations influenced by their personal perception of the risk from the floodwater or by the most popular destination selected by others.

These new developments are evaluated by analysing the associated changes induced in the simulated outcomes, by first contrasting them against the outcomes of the previous version of the simulator for a synthetic case study of a during-flood evacuation in a shopping centre; then, through a new real-world case study of a mass evacuation from the Hillsborough football stadium in response to a flood emergency replicating the conditions of November 2019 Sheffield floods.

This study is one step forward to developing an evacuation simulation tool, which intertwines an enhanced level of heterogeneity in agent characterisation and experimentally formulated behavioural rules for temporal and spatial microscopic flood risk analysis at individuals' level. …

■ *" There have been recent experimental and numerical studies which has shown improved HR assessment, e.g. Martinez-Gomariz et al. (2016). Why the authors used Equation quoted in line 133 for flood Hazard Ratio while they used more recent studies to evaluate flood HR? "*

We acknowledge that there have been other metrics proposed for assessment of flood risks on people in recent years. We too previously showed that HR could be overly optimistic when dealing with low-to-medium flood risks (see in Shirvani et al. 2020). But the focus of this study is to analyse the relative changes associated with the inclusion of different levels of sophistication in the behavioural rules and heterogeneity of pedestrian agents characteristics, not evaluation of different criteria for flood risk assessment. Considering this, applying the simulator for any other metric would potentially lead to similar trends of pedestrians' risk state. Also, since we have employed the TAPAS approach, it makes sense to reconsider the same risk criteria as before to enable a fair comparison with the previous observations.

More importantly, the risk to people has been analysed based on a combination of the HR quantity and the stability states of pedestrians, which are evaluated via experimentally-derived formulas incorporating both local floodwater flow variables and human-body characteristics. Sect. 2.1 has been shortened and revised to better explain the criteria employed to assess the flood risk to people in the present simulator including the stability states.

Changes made to the 2nd and 3rd paragraphs of Sect. 2.1:

… Each navigation agent is set to store the updated state of floodwater variables from the coincident flood agent and subsequently provide this information to the pedestrian agent(s) at their location. The recipient pedestrian agents use the flood information to change their states based on a self-evaluative assessment of two criteria: Hazard Rating (HR) quantity of floodwater and human-body stability limits.

The HR quantity in pluvial or fluvial flooding with low probability of debris could be estimated as HR = $(V + 0.5) \times h$ where $V$ stands for the velocity magnitude estimated as $V = \sqrt{u^2 + v^2}$ (Ramsbottom et al. 2006, Kvočka et al., 2016). Depending on the categorisation of the HR by the UK Environment Agency (EA), pedestrian agents are set to autonomously flag themselves with one of the four flood risk states: 'low' (0.0 < HR < 0.75), 'medium' (0.75 < HR <

1.5), 'high' (1.5 < HR < 2.5) and 'highest' (2.5 < HR < 20). In a similar way, the pedestrian agents are assigned a stability state which is also indicative of their mobility or immobility inside the floodwater. The stability state of pedestrian agents is estimated based on two experimentally derived formulas reported in Xia et al. 2014. These formulas evaluate the incipient velocity limits ($U_c$) for toppling and/or sliding conditions of human subjects in the floodwater by weighing the body height and mass information of each pedestrian agent as well as the states of floodwater variables. Depending on the evaluated $U_c$ and the magnitude of the floodwater velocity ($V$), the pedestrian agents are assigned one of four stability states: 'stable condition' where they carry on moving, or otherwise immobilised under 'toppling-only condition', 'sliding-only condition' or 'toppling-and-sliding condition' (see Shirvani et al. 2020 for more information).

- *Flood Hazard Ratio introduced in this study (line 133) also include a debris factor which has a significant impact on the HR. This is missing from the equation. what is the significance of removing this from the equation?*

The equation of the flood Hazard Rate (HR) quantity has a debris factor that is selected "*depending on the probability that debris will lead to a significantly greater hazard*" (Ramsbottom et al. 2006). As our simulator has been applied to study responses of people to a pluvial flood in small urban hubs, it makes sense to assume a low debris probability, leading to a zero debris factor based on UK Environment Agency's (EA) recommendation (Ramsbottom et al., 2006) and other academic studies (e.g. Willis et al., 2019). As shown in the text box above, the 3rd paragraph of Sect. 2.1 has been revised to clarify the choice for the debris factor: "*The HR quantity in pluvial/fluvial flooding with low probability of debris could be estimated as ...*"

- *In some places the paper is confusing. In particular, section 2 where there are many references to the other sections.*

As clarified in our reply to Referee #2, who also raised this point, Sect. 2 has been reduced, re-written and re-organised in a more concise manner to be more focused on the novelties proposed for the enhanced simulator. Note also that Sect. 2.3 in the initially submitted preprint) is now treated as a separate Sect. 3, which is dedicated to the evaluation and analysis of the new functionalities and properties added to enhance the simulator.

- *Justification to use Mode 2 with the simulator to plan evacuation case study involving severe flood is based on the lack of major changes as a result of implementation of Mode 4 (line 394). This seems to be insignificant and based on 1 simple case study. The reviewer expects a more detailed analysis for such decisions.*

New revisions include a sensitivity analysis on the averages obtained from the outputs a series of 20 runs for each of the five simulation modes, to account for the inherent variability randomness and stochasticity in the pedestrian agent characteristics rules implemented in the social force model (see also our reply to major comment #1 raised by Referee #1). The new results suggest that running the simulator with age-related moving speeds and with the two-way interaction condition (Mode 2 or Mode 4) is a sensible choice to study the stability state of large crowds in floodwater imposing low-to-medium risks to pedestrians. As the floodwater for the real-world case study has been identified to be dominated by low-to-medium flood HR, using Mode 2 to average out the probabilistic outputs is the most sensible choice. That being said, running the simulator under Mode 4 can be used but would only result in a faster evacuation speed. This option has not been investigated as it does not comply with the need to have a more conservative estimation for evacuation planning for a crowd with a heterogeneous population. The analysis in Sect. 3.3 has been revised to support the rationale behind choosing Mode 2 for the real-world case study.

The revised Sect. 3.3 now more clearly leads to the conclusion highlighted in the last two paragraphs, which is:

… Contrasting the predicted times reinforces previous findings from Fig. 4: compared to Mode 0, the age-related walking speeds, either with or without the two-way interaction condition (Mode 1 and Mode 2, respectively), leads to slower evacuation speed predictions that become faster under the running condition.

> Next, the simulator will be applied to analyse a scenario of mass evacuation of pedestrians during a pluvial flood leading to low-to-medium risk floodwaters in an urban neighbourhood. Supported by the analysis in Sect. 3.2 and Sect. 3.3, the simulator's configuration will be based on Mode 2 to produce conservative estimations of the evacuation time for planning and decision making.

■ *The authors provided a good literature review of the topic. However, their methodology and novelty only compare with their previous study, e.g. use of BMI (line 170), age and gender (line 10) are considered as novelty when compared to Shirvani et al. (2020). However, these have been used in flood hazard assessment previously.*

Answered in our previous replies to the 1st and 2nd comments brought by Referee #3.

■ *The paper seems too long in reviewer's view. It can be more concise by shortening some of the text which are less relevant to the novelty of this paper. Some of the figures and tables are also reproduced from other publications, e.g. Figure 2 and Table 1 and 2. Removing those and just citing them seems more appropriate. This will reemphasise the point about the overlaps between this and other publications.*

As also stated previously, Sect. 2 has been revised and made more concise, by also removing Table 1 and Table 2, to mostly focus on the new features added to enhance the simulator. However, we had to keep Fig. 2 as a minimum to help keep the paper self contained.

■ *It seems the characteristics are assigned randomly to the agents (line 292). What is the significance of such random assignment? For instance, how repeatable the simulations will be if they are repeated and what is the significance of this repeatability.*

Answered in our previous reply to Referee #1's major comment #1.

■ Part 1: *What is the significance of including conditions from both toppling-only and toppling-and- sliding (paragraph starting line 402)?*

We wish to clarify that the simulator is already featured with the capability to provide information on toppling-only, sliding-only and toppling-and-sliding conditions, in order to inform on the different potential risks of instability should a pedestrian be exposed to different ranges of floodwater depth and velocities. However, the simulation outputs for the shopping centre test case suggest that no pedestrians could be under a sliding-only condition for the type of flood used in this paper, which makes sense as this stability state is more likely to happen in shallowly fast flowing flash floods.

> This aspect has been pointed out as a footnote in Sect. 3.2:

> Although the sliding-only is implemented in the simulator, it is not expected to predict pedestrians under this stability state for the type of fluvial or pluvial floods investigated in this paper. This stability state would occur when pedestrians respond to raging and shallowly propagating floodwaters such as the case of a flash flood.

> Also, it was clarified in the discussion of Fig. 5 in Sect. 3.3:

> ...
> Figure 5 shows the trends in the number of evacuating pedestrians with different stability states averegated from the simulator predictions after 20 runs for all the configuration modes (Table 4). Pedestrians seem to be only under either toppling-only condition (Fig. 5a) or toppling-and-sliding condition (Fig. 5b), with no pedestrians spotted to be under a sliding-only condition. The trends predicted with the simulator under Mode 1 to Mode 4 lead to a …

Part 2: *Furthermore, a case study considered a shopping centre where the floor material expected to be different to the conditions where stability experiments were conducted. Furthermore, the slope and the direction of walking has impact on stability. The authors are expected to consider this somehow.*

The empirical formulas used to evaluate human stability, though may have been derived under limited laboratory conditions, requires as inputs body characteristics of a person and the floodwater variables. To the best of our knowledge, no other formulas exist to evaluate stability states considering both the human-body and the floodwater characteristics to support such computer-based simulation tools. In this setting, the impact of the slope and the material of the ground are inherent to prediction of the floodwater characteristics to, in turn, potentially influence the stability of individuals.

■ *Similarity between the predicted flood inundation levels and those of the EA flood maps were used as the validation of the model. It is stated that "the generated inflow hydrograph is able to replicate realistic floodwater depths and extents within the walkable area" (line 531). It is important to ensure that both simulations have the same conditions to be able to draw such a conclusion. For instance, is the EA flood maps produced under similar hydrographs? Furthermore, velocity is as important if not more important than the water levels in case of evacuation. It is important to show the velocities across the domain and provide reader with an assessment of uncertainties associated with such predictions.*

Answered in our reply to minor comment #5 raised by Referee #1 (see there, in particular the revisions applied to Sects. 4.1 and 4.2). In addition, as advised by the referee, we also included analysis of velocity maps alongside the flood depth maps reported by the EA to ensure that the simulated floodwater characteristics are fairly within the expected ranges.

*Minor comments:*

■ *What was the reason behind the orientation of the model used in this study? Was it not more efficient to have y-axis parallel to A61?*

The model is based on a north-up orientation to make it easier for the reader to spot the other three cardinal directions. It should be noted that the efficiency of the hydrodynamic model implemented in the flood-pedestrian simulator is neither related to nor driven by the orientation of the map.

■ *Line 152: what is nM?*

It is Manning's roughness parameter as stated in the 2nd paragraph of Sect. 2.1: "*A flood agent stores information of the terrain properties in terms of height (z) and Manning's roughness parameter ($n_M$); and the state of floodwater variables in terms of depth (h) and velocity components (u and v).*"

■ *Equation 2: where is the source of the equation and calculation of M? It might be better to be cited in the same way as Equation 1.*

The formula for specific force per unit width (*M*) is already explained below Eq. (2) in the 3rd paragraph of Sect. 2.2.2. We prefer using an in-line formula for *M* to keep the focus on outlining behavioural rules.

■ *It will be useful to also include whether the discharge was added only as mass or there is also momentum (velocity) associated with the flow.ents:*

As the simulator aims from a hydrodynamic model formulated based on a shallow water equation (SWE), it describes flow dynamics in terms of both mass and momentum; thus, the inflow was introduced at the boundary as discharge. In the revised paper, as stated in our reply to minor comment #3 of Referee #1, a sentence has been added to clarify that *"The hydrodynamic model is formulated based on a non-sequential implementation of a finite volume solver of the depth-averaged shallow water equations on a two-dimensional grid...".*